# Transversus Abdominis Plane Block as a Strategy for Effective Pain Management in Patients with Pain during Laparoscopic Cholecystectomy: A Systematic Review

**DOI:** 10.3390/jcm11236896

**Published:** 2022-11-22

**Authors:** Abdalkarem Fedgash Alsharari, Faud Hamdi Abuadas, Yaser Salman Alnassrallah, Dauda Salihu

**Affiliations:** 1College of Nursing, Jouf University, Sakaka 72388, Saudi Arabia; 2Ministry of Health, Riyadh 12613, Saudi Arabia

**Keywords:** cholecystectomy, laparoscopy, pain management, postoperative, anesthesia

## Abstract

Laparoscopic cholecystectomy (LC), unlike laparotomy, is an invasive surgical procedure, and some patients report mild to moderate pain after surgery. Transversus abdominis plane (TAP) block has been shown to be an appropriate method for postoperative analgesia in patients undergoing abdominal surgery. However, there have been few studies on the efficacy of TAP block after LC surgery, with unclear information on the optimal dose, long-term effects, and clinical significance, and the analgesic efficacy of various procedures, hence the need for this review. Five electronic databases (PubMed, Academic Search Premier, Web of Science, CINAHL, and Cochrane Library) were searched for eligible studies published from inception to the present. Post-mean and standard deviation values for pain assessed were extracted, and mean changes per group were calculated. Clinical significance was determined using the distribution-based approach. Four different local anesthetics (Bupivacaine, Ropivacaine, Lidocaine, and Levobupivacaine) were used at varying concentrations from 0.2% to 0.375%. Ten different drug solutions (i.e., esmolol, Dexamethasone, Magnesium Sulfate, Ketorolac, Oxycodone, Epinephrine, Sufentanil, Tropisetron, normal saline, and Dexmedetomidine) were used as adjuvants. The optimal dose of local anesthetics for LC could be 20 mL with 0.4 mL/kg for port infiltration. Various TAP procedures such as ultrasound-guided transversus abdominis plane (US-TAP) block and other strategies have been shown to be used for pain management in LC; however, TAP blockade procedures were reported to be the most effective method for analgesia compared with general anesthesia and port infiltration. Instead of 0.25% Bupivacaine, 1% Pethidine could be used for the TAP block procedures. Multimodal analgesia could be another strategy for pain management. Analgesia with TAP blockade decreases opioid consumption significantly and provides effective analgesia. Further studies should identify the long-term effects of different TAP block procedures.

## 1. Introduction

Laparoscopic cholecystectomy (LC) is a minimally invasive technique that causes mild postoperative discomfort in the parietal, visceral, incisional, and referred regions [1]. In these patients, multimodal approaches [2], epidural analgesia, and intraperitoneal injection of local anesthetics (LA) are often used in conjunction with patient-controlled intravenous analgesia. Transversus abdominal plane (TAP) block is a well-known procedure for postoperative analgesia during laparoscopic abdominal surgery as part of this approach [3]. TAP block is safe; it reduces or eliminates the need for analgesics and has fewer side effects such as postoperative nausea and vomiting (PONV) [4]. In addition, several physicians are actively improving the precision of LA absorption by ultrasound [5,6,7,8]. Thus, this innovative approach has demonstrated the analgesic efficacy of laparotomy and laparoscopic procedures [9].

Rafi [10] pioneered the TAP block in 2001 as a historically guided practice for achieving a field block through the petit triangle. In this procedure, a solution (LA) is injected further into the plane between the obliquus internus and transversus abdominis muscles. The thoracolumbar nerves travel through this plane after exiting the T6 to L1 spinal roots, directing sensory nerves to the anterolateral abdominal wall [11]. The propagation of LA in this plane blocks neurological afferents and provides analgesia to the anterolateral abdominal cavity. TAP blockades are becoming technically easier and more feasible as ultrasound technology advances. As a result, curiosity about TAP blocks as a clinical tool for analgesia after abdominal surgical treatment has increased. The most commonly reported pain during laparoscopic cholecystectomy was of moderate to severe intensity [12].

TAP blocks are effective for a number of abdominal practices, including hysterectomy, cesarean section, cholecystectomy, colectomy, hernia repair, and prostatectomy [10,13,14,15]. Since the analgesic effect is limited to somatic pain and has a short life span [16], a single TAP blockade is efficient in multimodal analgesia. TAP blockades could solve the problem of limited duration by continuous infusion [17,18] or prolonged release of liposome’s LA [19]. In contrast, clinical studies on TAP block yielded negative results [20,21]. Consequently, analgesic consistency, duration of analgesia, patient comfort, and different corporate strategies need further analysis. Numerous regional anesthetic adjutants such as Dexmedetomidine, Clonidine, Epinephrine, and Dexamethasone are usually combined with enhancement of analgesic efficacy and length chains [22,23]. 

Most patients undergoing laparoscopic cholecystectomy experience pain in the first 24 h after surgery, with port sites being the most painful. After laparoscopic surgery, pain is mainly felt as visceral pain due to the trauma of gallbladder resection and parietal pain due to skin incision [24]. However, the frequency and intensity of incisional pain were higher than visceral pain after laparoscopic cholecystectomy. Therefore, to optimize postoperative pain control in these patients, analgesic studies should focus on reducing incisional pain.

A number of reviews have been conducted on postoperative pain management, some of which include ultrasound-guided transversus abdominis pain block, with some exploring the best anesthetic technique [25,26,27,28] and abdominal surgeries [3,29,30,31,32,33,34]. Other reviews focused on specific conditions, such as colorectal surgery [35,36], wound infiltration [33,37], caesarean delivery [38], bariatric surgery [39,40], lower abdominal incisions [16], breast reconstruction [41], and minimally invasive surgery [42]. Notably, few studies investigated laparoscopic cholecystectomy. Koo, Hwang, Shin, and Ryu [43] investigated the use of an erector spinae block in patients undergoing laparoscopic cholecystectomy. Ni, Zhao, Li, Li, and Liu [44] and Zhao et al. [45] studied the effects of transversus abdominis block on laparoscopic cholecystectomy, while Peng et al. [28] studied the efficacy of ultrasound-guided laparoscopic cholecystectomy. Specifically, some studies examined the clinical safety and efficacy results of the TAP block across clinical domains [3,46,47,48,49] or identifying the best evidence [50]. There appears to be a paucity of data on the optimal dose of TAP block anesthesia for laparoscopic cholecystectomy and the procedure’s long-term effects [28]. Again, there seems to be no attempt to comprehensively compare the analgesic efficacy of different TAP strategies for laparoscopic cholecystectomy. Furthermore, the clinical significance of TAP remains unclear and no study compares the types of anesthetic agents used and their dosages, hence the need for this study.

## 2. Objectives

This review should therefore achieve the following objectives:Explore the optimal dose of TAP block anesthesia for laparoscopic cholecystectomy.Identify the types and concentrations of local anesthetics and other supportive agents commonly used for laparoscopic cholecystectomy.Compare the analgesic efficacy of different types of TAP block procedures and their long-term effects.Examine the clinical significance of TAP.

## 3. Methods

This review was prepared in accordance with the 2020 PRISMA (Preferred Reporting Items for Systematic Review and Meta-analysis) guidelines [51]. 

## 4. Eligibility Criteria

### Inclusion Criteria

5.Population: Adult patients undergoing LC.6.Intervention: Postoperative pain management using TAP block or in combination with adjutants.7.Comparators: Active placebo or adjunct treatment.8.Outcomes: Postoperative use of analgesia use if they reported a visual analogue scale (VAS) or numeric rating scale (NRS) outcome of postoperative pain after 24 h.9.Design: Randomized trials published in peer-reviewed journals.

## 5. Exclusion Criteria

Articles were considered ineligible if they were letters to the editor, commentary, not peer-reviewed papers (e.g., dissertations), case studies, were not published in full text (e.g., conference proceedings), or were non-experimental studies (e.g., qualitative studies). Furthermore, studies that did not include an active control group were also excluded. 

## 6. Information Sources

The five electronic databases of PubMed, Academic Search Premier, Web of Science, CINAHL, and Cochrane Library were searched for randomized trials. The search was conducted primarily between 12 and 13 August 2022. A supplemental search was conducted in Google Scholar and trial registries (http://clinicaltrials.gov/, accessed on 15 August 2022). All registries and conclusions with the keywords “TAP block laparoscopic cholecystectomy”, “postoperative pain in laparoscopic cholecystectomy”, and “postoperative pain management in hospitalized patients undergoing laparoscopic cholecystectomy” were explored. 

## 7. Search

We employed the keywords (search terms) transversus abdominis block OR transversus abdominal plane OR transversus abdominis plane AND laparoscopic cholecystectomy OR laparotomy. We limited the search to published articles from the beginning to the present. We considered articles that were published in English and had an abstract. The publication period ranged from inception to the present day. We also manually searched the reference lists for eligible studies.

## 8. Study Selection

Studies were selected using the PRISMA framework [51]. Results from the five databases were exported to Endnote Reference Manager, and duplicates were removed. The titles and abstracts were screened according to the eligibility criteria. Eligible studies for inclusion were identified after the full-text screening. Two independent reviewers (A.F.A. and D.S.) performed the screening independently, and a third reviewer (F.H.A.) was asked to clarify any discrepancies identified during the process.

## 9. Data Collection Process

Data were extracted from eligible studies and entered into a spreadsheet in Microsoft Excel. Two independent reviewers performed this task. A third reviewer was asked to clarify any discrepancies. The authors of the papers under consideration were contacted for queries or clarifications.

## 10. Data Items

Information on the study profile included participant demographics (i.e., gender and age), study design, population characteristics, sample size, study groups (intervention and control), and duration of follow-up. Components of the intervention include the setting in which it was delivered, the dose (frequency, duration, and course) of TAP, the total exposure in minutes to TAP, and the associated theories underlying the therapeutic effects of TAP as reported. For the outcomes, we extracted the mean (M) and standard deviation (SD) for VAS, NRS, and postoperative pain for all groups (i.e., intervention and control) at baseline, post-intervention, and follow-up. For accuracy, two reviewers (A.F.A. and D.S.) performed this task.

## 11. Risk of Bias in Individual Studies 

Appraisals of the eligible studies were conducted using the Physiotherapy Evidence Database scale (PEDro scale). Because of its high construct validity, the PEDro scale was selected for evaluation in randomized controlled trials [52]. To obtain a PEDro total score, items 2–11 were summed. The Internal Validity subscale is scored with items 2–9, and the Statistical Reporting subscale is scored with items 10 and 11 [53,54]. The study is classified as moderate if it scores 4–5, good if it scores 6–8, and excellent if it scores 9–10 [55].

## 12. Summary of Measure and Synthesis of Results 

The Cochrane Handbook for Systematic Reviews of Interventions was used as a guide for data processing [56]. A meta-analysis would be conducted in cases where more than two studies were eligible and measured the same outcomes at which data might be collected: T0 (baseline), T1 (immediately after intervention), and T2 (follow-up). We used a standardized approach in reporting results because different studies measured the same outcome differently. Given the differences in group means and variance within the study population, we determined the minimum clinically significant difference (MCID) using the distribution-based approach [57]. A small effect was described as 0.2, a medium effect as 0.5, and a large effect as 0.8, based on Cohen’s d [58].

## 13. Results

As shown in Figure 1, a total of 758 results were generated from the five databases: CINAHL (*n* = 37), PubMed (*n* = 154), Web of Science (*n* = 183), Cochrane (*n* = 196), and Academic Search Premier (*n* = 183). Manual searches yielded five results (*n* = 5). After deduplicating 231 papers with endnotes and manual search, 527 papers were used for the title and/or abstract screening. After title and/or abstract screening, 77 records were selected for full-text screening. Twenty-nine records were removed after full-text screening for failure to meet the criteria. Forty-eight articles were included in the qualitative and quantitative synthesis.

## 14. Risk of Bias within Studies

As shown in Table 1, nineteen (39.6%) did not conceal the assignment of participants. Thirteen (27.1%), twenty-nine (60.52%), and thirty-one (64.58%) studies did not blind assessors, participants, and therapists, respectively. Twenty-six studies underwent an intention-to-treat analysis. In all studies, the dropout rate was less than 15%. All studies reported between-group statistical comparisons, point measures, and variability data. Overall PEDro ratings ranged from 4 to 10, with only one study rated as excellent and three as moderate, while the rest were of good quality.

## 15. Study Characteristics

As shown in Table 2, a total of 3651 subjects participated in the study, with the majority being female (*n* = 1822, 49.9%) and ages ranging from 18 to 80 years. The TAP block techniques ranges from ultrasound-guided transversus abdominis plane block, oblique subcostal transversus abdominis plane, posterior transversus abdominis plane block, erector spinae plane, subcostal Transversus abdominis, and transversus abdominis plane block. Other approaches were transversus abdominis plane block, oblique subcostal transversus abdominis, subcostal transversus abdominis, subcostal block, subcostal transversus abdominis plane block, quadratus lumborum, laparoscopic transversus abdominis plane, rectus sheath block, blocking the branches of intercostal nerves at the level of mid-axillary line, and laparoscopic subcostal TAP. Infiltration of the surgical site was 10 mL, 15 mL, 16 mL, 20 mL, 30 mL, 40 mL, and 100 mLand either unilateral or bilateral or to the respective or conventional port sites using local anesthetics or other relevant agents. Eighteen (37.5%) of the studies reported the use of patient-controlled analgesia (PCA) or patient-controlled intravenous analgesia (PCIA). The Visual Analogue Scale is the most commonly used outcome measure (*n* = 30, 62.5%).

As shown in Table 2, seven classes of drugs were used as premedication before induction of anesthesia: benzodiazepines (Lorazepam 2 mg; Midazolam 0.12 mg/kg, 1–2 mg, 0.01–0.02 mg/kg, 0.03 mg, 0.05 mg/kg, 0.5 mg and 7.5 mg; Diazepam 10 mg; Alprazolam 0.25 mg/kg), analgesics (Paracetamol 15–20 mg/kg; Diclofenac 0.5 mg/kg; Fentanyl 20 mcg, 2 mcg/kg; Etocoxib 120 mg), prokinetic agents (Metoclopramide 10 mg), anticholinergics agents (Glycopyrrolate 0.003 mg/kg, 0.05 mg/kg, 0.2 mg), 5-HT3 antagonists (Ondansetron 4 mg), H2 receptor blockers (Ranitidine 150 mg), and alkalizing agents (Ringer lactate solution 500 mL).

### 15.1. Objective #1. The Optimal Dose of TAP Block Anesthesia for Laparoscopic Cholecystectomy

As already shown in Table 2, the minimum dose of local anesthetic for transversus abdominis plane blockade could be 20 mL for US-TAP and US-OSTAP and 0.4 mg/kg for port infiltration. However, the agents and their concentration may vary.

### 15.2. Objective #2. Types and Concentrations of Local Anesthetics and Other Supportive Agents Commonly Used for Laparoscopic Cholecystectomy

As shown in Table 2, the most commonly used solutions were Ropivacaine (0.2%, 0.25%, 0.4%, 0.5%, 0.75%, 0.365%, and 0.375%), Bupivacaine (0.25%, 0.5%, and 0.375%), Levobupivacaine (0.25%, 0.5%, and 0.375%), and Lidocaine (2%, 5 mg). Other supportive agents that can be used alone or in addition to the above anesthetics are 0.9% Normal saline (1 mL,2 mL, 10 mL, 20 mL, 32 mL, 40 mL, 100 mL), Dexamethasone (2 mL,1 mcg/kg, 4 mg), Magnesium Sulfate (0.5 mg), Oxycodone (40 mg), Ketorolac (180 mg), Dexmedetomidine (0.5 mcg/2 mL), Epinephrine (5 mcg/mL), Esmolol 0.05 mg/kg, Tropisetron (10 mg), and Sufentanil (2 mg/kg). The most commonly used local anesthetic is Bupivacaine (*n* = 27, 5.3%), followed by Ropivacaine (*n* = 17, 35.4) in different concentrations. From this review, US-TAP block is the most commonly used procedure (*n* = 31), followed by port infiltration (*n* = 13), general anesthesia only (*n* = 13), and US-OSTAP blocks (*n* = 7).

### 15.3. Objective #3. Effects of TAP Block Anesthesia

a.Comparison of different approaches to ultrasound-guided blockade of the transversus abdominis plane

#### 15.3.1. Analgesic Efficacy of Ultrasound-Guided Transversus Abdominis Plane Block Versus General Anesthesia Only

As shown in Table 3, Morphine consumption was greater in the control compared to the intervention group. Notably, the mean change in the intervention group from the start to the end of laparoscopic cholecystectomy was 9.6, *p* < 0.005, and the control group had a mean change of 20.5, *p* < 0.005 [59]. Between the first 20 min and 24 h postoperative, the two US-TAP groups using 0.25% and 0.5% Levobupivacaine reported lower postoperative pain scores of 3.3 to 1.6 and 3.2 to 1.3, respectively, compared with those receiving anesthesia alone (8.6 to 4.4) [7]. In addition, analgesic requirements between 20 min and 12 h postoperative were higher in the control group (−4.2 to −0.7) than in patients receiving 0.25% Levobupivacaine (−1.7 to −0.5) and 0.5% Levobupivacaine (−1.9 to −0.5) [7]. Similarly, the control group appeared to have higher pain scores between 0 and 24 h postop (2.35 to 1.3) compared to those who received posterior US-TAP block (1.2 to 0.8) and subcostal US-TAP block (0.85 to 0.15) [6]. Analgesic requirement between 0 and 12 h is also higher in the control group (−1.0 to −0.6) than in the posterior US-TAP block (−0.4 to −1.0) and subcostal US-TAP block (−0.7 to −0.65) [6]. Moreover, pain perception between 10 min and 24 h postoperative was higher in the control group at rest (6.6 to 2.1) than in the US-TAP group (4.2 to 2.1) and during cough (7.5 to 3.0 and 4.7 to 2.9, respectively [61]. The need for analgesics between 10 min and 6 h was also higher in the control group (−4.5 to −1.2) than in US-TAP (−1.8 to −1.8) [61]. 

As shown in Appendix A, the most commonly usedintra-operative opioid analgesic was Fentanyl (*n* = 27), followed by Remifentanil (*n* = 10). Intra-operative opioids were switched to Tramadol (*n* = 14), Morphine (*n* = 12), Ketorolac (*n* = 5), and Nalbuphine (*n* = 1). However, in some cases, Diclofenac sodium (*n* = 6) and acetaminophen (*n* = 1) were also used.

Opioid consumption was found to be lower in those receiving US-TAP block with 0.25% Bupivacaine 2.1(0.5) and Bupivacaine 0.25% plus Magnesium Sulfate 0.5 g, 2.2(0.5) compared with those receiving general anesthesia alone 2.8(0.6) [65]. Compared with the control group, the first need for rescue analgesics was delayed in the US-TAP and US-TAP (Dexamethasone) groups (403.0, 436.0 versus 152.3, *p* < 0.01). Patients in the US-TAP and US-TAP (perineural Dexamethasone) groups had lower pain scores on the numeric rating scale (*p* > 0.01) and used fewer postoperative analgesics (*p* < 0.01) [67]. The need for analgesics was higher in the control group than in the group US-TAP [68]. The number of patients requiring additional analgesics (Morphine and Ketorolac) was higher in the group of patients receiving US-TAP block with in situ indwelling catheters than in the control group and the group of patients receiving US-TAP block + PCA [71]. There was a significant reduction in pain perception with US-TAP block (4.4 to 1.4) when compared with the control group (6.4 to 2.2) [96]. Again, the mean analgesic requirement between the first 2 and 24 h after surgery was found to be higher in the control group (−42 to −1.0) than in the US-TAP group (−3.0 to −0.5) [96].

Comparing the unilateral and bilateral US-TAP block with the control group, both strategies appear to be effective. Pain perception at rest and during coughing was lower with the unilateral (right) US-TAP (3.10 to 1.23 and 3.37 to 1.8, respectively) and the bilateral US-TAP block (3.33 to 1.23 and 3.47 to 1.57, respectively), compared with the control group (6.03 to 2.10 and 6.33 to 3.07) [100]. Analgesic requirements between 1 and 12 h postoperative both at rest and during cough were higher in the control group (−3.39 to −0.07 and −3.26 to −0.06, respectively) compared with US-TAP block (unilateral) (−1.87 to −0.17 and −1.57 to −0.2, respectively) and US-TAP block (bilateral) (−2.1 to −0.4 and −1.9 to −0.26, respectively) [100]. 

#### 15.3.2. Analgesic Efficacy of Ultrasound-Guided Transversus Abdominis Plane Block versus Port SITES Infiltration 

Between the onset of surgery and 24 h after surgery, the demand for opioids (Morphine) was slightly higher in the US-TAP block group (14.6) than in the port infiltration group (14.4) [20]. This demand pattern continued after 24 h with 11.1 for the US-STAP block group and 9.6 for the port group [20]. However, there seems to be no difference in the need for hydrocodone [20]. The port infiltration group appears to have a higher need for diclofenac (0.65), Paracetamol (21.0), and Fentanyl (3.1) compared with the US-STA block group (diclofenac (0.58), Paracetamol (16.0), and Fentanyl (3.0) [60]. In the US-TAP group and the local infiltration group, the 24 h Morphine requirements (mean) were 34.57 mg and 32.76 mg, respectively (*p* = 0.688). A total of eight patients in the US-TAP group and 16 patients in the local infiltration group required additional fentanyl intra-operatively (*p* = 0.028). In the immediate postoperative period, local infiltration levels were significantly higher at rest and during cough (*p* = 0.034 and *p* = 0.007, respectively) [66]. Patients who received subcostal US-TAP block had a statistically significant reduction in postoperative pain within the first 24 h after surgery compared with port infiltration. In the subcostal TAP block group, opioid consumption was lower over 24 h (125 mg versus 175 mg *p* < 0.001). US-TAP block and local infiltration had significantly different postoperative pain scores (VAS) at 0, 2, 6, 12, and 24 h [101]. The VAS score is significantly higher in the local infiltration group than in the US-TAP block group (*p* < 0.001) [101].

Patients receiving the subcostal block US-TAP had a greater delay in requesting rescue analgesics (3.20 versus. 1.70, *p* < 0.001) [73]. Time to first analgesic (mean) was 292.7 and 510.3 min in the portsite infiltration and subcostal US-TAP block groups, respectively, and mean Tramadol requirements were 141.8 mg and 48.69 mg (*p* = 0.001 for both) [76]. Mean NRS at 2, 3, 6, 12, and 24 h was significantly lower in the subcostal block US-TAP group (0.03, 0.43, 1.35, 0.93, and1.13) than in the port infiltration group (0.30, 2.05, 3.10, 2.48, and 2.25) [76]. Mean analgesic requirements at 2, 3, 6, and 12 h per group were higher in the port infiltration group (1.95, 0.2, −0.85, and −0.23) than in the US-TAP block group (1.1, 0.7, −0.22, and 0.2) [76]. Similarly, mean pain scores between 0 and 24 h postoperative were higher in the port infiltration group (4.42, 4.64, 5.08, 1.52, and 3.92) than in the unilateral US-TAP block group (2.65, 3.20, 3.47, 1.36, and 2.67) [80]. Likewise, analgesic requirements between 0 and 8 h were higher in the port infiltration group (−0.5, −0.72, −1.16, and 2.4) than in the US-TAP block group (0.02, −0.53, −0.8, and 1.31) [80]. There was a significant difference between the US-TAP block group and the local infiltration group at 10 min, 30 min, 1 h, and 2 h [2 (0–2.5) versus 0.011]; [1.5 (0–3) versus 3 (2–5); *p* = 0.001]; [1.5 (0–2) versus 2 (2–3); *p* = 0.001; and [2 (0–2) versus 2 (1.5–2.5); *p* = 0.010] [81]. Additionally, US-TAP block patients were significantly less likely to require intra-postoperative opioids and rescue analgesia (*n* = 5, *n* = 8, *p*< 0.001) compared to port site infiltration (*n* = 38, and 30, *p* < 0.001) [81]. Finally, the US-TAPB block group had significantly lower resting VAS values than the addition of 2 mg/kg Sufentanil or Nalbuphinevia PCA at 2, 6, 12, 24, and 48 h postop (*p* < 0.05) [98].

It is noteworthy that a similar pattern of changes, in which the US-TAP block group had a lower pain experience and analgesic requirement compared with the port infiltration groups, was observed in a study by Khandelwal, Parag, Singh, Anand, and Govil [82] and by Arık, Akkaya, Ozciftci, Alptekin, and Balas [86]. A similar pattern of changes was observed and the time to rescue analgesia was equally longer in patients receiving subcostal US-TAP block (3.63 h) compared to the port infiltration group (1.73 h, *p* = 0.0002) [87]. 

Uniquely, Liang, Chen, Zhu, and Zhou [88] used different concentrations of Ropivacaine (0.5% and 0.25%) plus 1 mcg/kg Dexamethasone in addition to preoperative administration of solution at the trocar site before surgery (group LAI), with additional US-TAP block in the TL group, while the TR group received rectus sheath block in addition to US-TAP block; however, the three groups did not differ in pain scores at any time point within the 48 h period.

#### 15.3.3. Analgesic Efficacy of Ultrasound-Guided Transversus Abdominis Plane Block before Induction of Anesthesia and after Surgery

Although there does not appear to be a significant difference between the two groups, mean meperidine consumption (mg) was higher in those receiving US-TAP block (after surgery) (34.21) than in those receiving US-TAP block (induction) (32.11) [102]. Postoperative pain perception in the first 30 min to 24 h was also higher in those who received a US-TAP block (post-surgery) (4.96 to 4.63) compared to US-TAP block (pre-induction) (3.18 to 3.47) [95]. Likewise, analgesic requirements in the first 30 min to 12 h after surgery were higher for the US-TAP block group (post-surgery) (−0.33 to −0.08) compared with the US-TAP block group (pre-induction) (0.29 to −0.06) [95]. On the contrary, time to first analgesic request was shorter for US-TAP blockade group (after surgery) (2.22, *p* = 0.089) than for US-TAP blockade group (induction) (5.80, *p* = 0.089) [102]. 

#### 15.3.4. Analgesic Efficacy of Ultrasound-Guided Transversus Abdominis Plane Block Using Different Concentrations of Local Anesthetics versus Normal Saline

We found that the US-TAP block group (saline) had higher 24hfentanyl consumption (877.8 mcg) than the US-TAP block group (0.5% bupivacaine + normal saline) (566.7 mcg) or the US-TAP block group (0.5% Bupivacaine + Sufentanil) (555.6 mcg; *p* = 0.03) [64]. Compared with the US-TAP (0.5% bupivacaine + normal saline) and US-TAP (0.5% Bupivacaine + Sufentanil) block groups, the postoperative pain score was higher with the US-TAP (saline) block group (*p* = 0.006); however, the intervention groups did not differ significantly [64]. Time to first fentanyl requirement was significantly less with US-TAP block (saline) (79.44) than with US-TAP block (0.5% Bupivacaine + Sufentanil) (206.38; *p* = 0.001) [64]. Intra-operative consumption of Remifentanil and postoperative VAS scores show that the US-TAP group (Bupivacaine plus 20 mL of normal saline) received a larger volume of local anesthetic solution, albeit at a lower concentration, and required fewer postoperative analgesics than the US-TAP group (Bupivacaine plus 10 mLof normal saline) [72]. The percentage of patients requiring Paracetamol (*p* < 0.002) and Nalbuphine (*p* < 0.001) as rescue analgesics was significantly lower in the US-TAP block (0.25% Bupivacaine) group (17.0, 68% and 2, 8.0%%) than in the US-TAP block (0.9% normal saline) group (25, 100%, and 24, 96%) [74]. In contrast, it has been found that Levobupivacaine 0.375% and 0.9% saline groups consumed similar amounts of opioids 24 h after surgery: 21.2 mg versus 25.2 oral Morphine equivalent; *p* = 0.48 [79]. Mean Morphine consumption after surgery in patients receiving US-TAP block (Esmolol) was 5.83 mg (*p* = 0.204) compared with US-TAP block (saline) (7.5 mg, *p* = 0.204) [89]. The US-TAP block (Esmolol) group had significantly lower early postoperative pain scores (*p* = 0.05) [89]. From arrival at PACU to 12 h postoperative, the mean analgesic requirement appears to be higher for the US-TAP block (saline) (2.5 to 0.1) than for the US-TAP block (Esmolol) (2.16 to −0.04) (Abdelfatah& Amin, 2021). The median Morphine consumption at 0–2 h postoperative was 7.5 mg for the US-TAP block (saline) compared with 5 mg for the US-TAP block (Ropivacaine) [21].

#### 15.3.5. Analgesic Efficacy of Ultrasound-Guided Transversus Abdominis Plane Block Using Different Concentrations of Local Anesthetics

It appears that the higher the concentration of anesthetics used for the US-TAP block, the lower the experience of pain. Of note, Ra et al. [7] used 0.25% and 0.5% Levobupivacaine differently; however, patients who received 0.5% had less pain in the first 20 min to 12 h after surgery. In contrast, the need for a analgesics in this group seems to be higher in the first 20 to 60 min after the procedure [7]. Interestingly, patients who received US-TAP block with 0.25% Levobupivacaine reported a high need for analgesia (−1.0) compared to those receiving 0.5% Levobupivacaine (−0.1) six hours later [7]. Similarly, the mean pain experience at 24 h was higher for those who received US-TAP block with 0.25% Levobupivacaine (1.6) compared to 0.5% Levobupivacaine (1.3) [7]. After 10, 30, and 60 min, patients receiving ultrasound-guided TAP blocks (0.375% Ropivacaine) had significantly lower pain scores than patients receiving US-TAP blocks (0.25% bupivacaine) [69]. The median [interquartile range] of postoperative analgesic requirements and cumulative rescue analgesic requirements were the same for both drugs (0.75% bupivacaine for US-TAP block versus 0.375% Ropivacaine for US-TAP block, *p* = 0.366) [69].

#### 15.3.6. Comparison of Analgesic Efficacy of Ultrasound-Guided Oblique Subcostal Transversus Abdominis Block versus Transversus Abdominis Plane Block

Because US-OSTAP is a relatively new technology that is essentially the same as TAP block procedures, we compared the analgesic efficacy of ultrasound-guided oblique subcostal transversus abdominis block with transversus abdominis plane block. Intra-operatively, there appears to be no difference between the US-TAP and US-OSTAP blocks in terms of Ketorolac consumption; however, the two groups differ in terms of intra-operative fentanyl consumption (US-TAP: 72.4 and US-OSTAP: 78.1) and postoperative Nalbuphine consumption (US-TAP: 7.3 and US-OSTAP: 8.0), with US-TAP block reporting less consumption [61]. In contrast, the US-TAP block group consumed more fentanyl postoperatively (US-TAP: 10.0 and US-OSTAP: 6.7) [61]. Mean postoperative pain scores between 10 min and 24 h at rest/coughing were also higher in the US-TAP group (4.3 to 2.1)/(4.7 to 2.9) than in the US-OSTAP (2.3 to 1.3)/(2.9 to 2.1) [61]. In addition, comparing the effects of the US-OSTAP procedure with only general anesthesia, the VAS pain scores at rest and upon movement were significantly lower in the OSTAP group on arrival at PACU and 2 h postoperative [62]. Total postoperative Tramadol requirements were significantly lower in the OSTAP group at 0–2 h (31.6) and 2–24 h (126.3) than in the control group at 0–2 h (80.3) and 2–24 h (267.1) [62]. In addition, a high proportion of patients who received US-OSTAP block (17, 85%) did not require additional analgesia compared with the US-TAP group (11, 55%) [68].

It should be noted that we compared US-OSTAP (normal saline) with US-OSTAP (Pethidine); however, US-OSTAP (Pethidine) significantly reduced pain scores at 0, 2, 4, 6, 12, and 24 h (*p* = 0.001) [70]. US-OSTAP (Pethidine) patients consumed significantly fewer opioids during surgery than US-OSTAP (normal saline) (150 versus 400 mg, *p* = 0.001), and opioid consumption during the first 24 h was significantly lower (20.4 versus 78 mg, *p* = 0.001) [70]. There were statistically significant differences between the US-OSTAP (bupivacaine) and US-OSTAP (Pethidine) groups when comparing VAS scores of US-OSTAP (bupivacaine) and US-OSTAP (Pethidine) (0 h) for pain intensity [70]. Therefore, as an alternative to 0.25% bupivacaine, 1% Pethidine might be used to achieve OSTAP blockade during laparoscopic cholecystectomy [70]. Pain experience was higher with US-OSTAP (normal saline) blockade from 0 to 24 h (2.21 to 1.52) than with US-OSTAP (Ropivacaine) blockade (0.71 to 0.38) [83]. Similarly, analgesic requirements were equally higher in the US-OSTAP (normal saline) block group (−0.69 to −1.19) compared with the US-OSTAP (Ropivacaine) block group (−0.33 to 0.18) from 0 to 6 h postoperative [83]. 

The US-OSTAP block (Ropivacaine) had a low fentanyl consumption of 122 mcg intra-operatively compared to US-OSTAP (normal saline) block (126.19 mcg [83]. Opioid consumption at PACU within the first 8 h after surgery was higher for the US-OSTAP (normal saline) block (9.52 mg) compared with the US-OSTAP (Ropivacaine) block (4.64 mg) [83]. In contrast, the US-OSTAP (Ropivacaine) block appears to have a higher opioid requirement between 8 and 16 h but not after 24 h [83].

#### 15.3.7. Analgesic Efficacy of Ultrasound-Guided Transversus Abdominis Plane Blockade with Equal Anesthetic Concentration and Adjuvant of Dexamethasone, and Dexmedetomidine versus Normal Saline

Postoperatively, it took 485.6 min for the first analgesic to be requested in US-TAP block (Dexmedetomidine), compared with 289.8 min in US-TAP block (normal saline) [75]. Patients in the US-TAP block group (Dexmedetomidine) consumed less cumulative Morphine in the first 24 h than patients in the US-TAP block group (normal saline) [75].

#### 15.3.8. Analgesic Efficacy of Ultrasound-Guided Subcostal Transversus Abdominis Plane versus Quadratus Lumborum Blocks

The time for first analgesic consumption using the US-TAP block was 63.72 min compared with the quadratus lumborum block at 70.00 min [78]. Similarly, Tramadol consumption (mg) was higher in the quadratus lumborum 86.66 mg than in the US-TAP block 83.43 [78]. Using the VAS scale, with the exception of 12 h postoperative, pain perception seems to be higher in the US-TAP block group (1.33 to 0.47) compared to quadratus lumborum (1.03 to 0.42) from 0 to 6 h [78]. However, this seems to differ in 12 and 24 h for the US-TAP block group (0.20 and 0.11) compared to the quadratus lumborum (0.22 to 0.09) [78]. In other words, pain perception using the DVAS scale appeared to be the same at 0 h but was higher in the US-TAP block group at 1 h (1.94) and at 12 h (1.00) [78]. In contrast, this pattern of change was observed in the quadratus lumborum after 6 h (1.50) and after 12 h (0.46) [78]. 

#### 15.3.9. Analgesic Efficacy of Ultrasound-Guided Transversus Abdominis Versus Quadratus Lumborum Blocks

The analgesic request was found to be higher among those receiving the US-TAP block (18, 72%) as compared to the quadratus lumborum block (14, 56%). Cumulative daily Morphine consumption was significantly higher in the US-TAP block group (6 mg (6–9)) than in the quadratus lumborum block group (3 mg (3–6)), *p* = 0.001 [97]. The median time to first analgesic request was longer in the quadratus lumborum block group (17 h (12, 24)) than in the US-TAP block group (8 h (6, 24)), *p* ≤0.001 [97].

b.Comparison of different approaches to laparoscopic-assisted transversus abdominal blockade

#### 15.3.10. Comparison of Analgesic Efficacy Analgesic Efficacy of Laparoscopic-Assisted Transversus Abdominal Block with Bupivacaine versus Normal Saline

In the LAP-TAP group (Bupivacaine + periportal saline injection), numerical pain assessment scores were significantly decreased after 1, 3, and 6 h of rest (*p* = 0.025, *p* = 0.03, and *p* = 0.007, respectively) as compared to normal saline [63]. 

#### 15.3.11. Analgesic Efficacy of Laparoscopic-Assisted Transversus Abdominal Block versus Port Sites Infiltration

Compared to port site infiltration, there seems to be no change in LAP-TAP block for pain at rest between the first 3 and 6 h postoperative [94]. However, it appears that the port infiltration group had more pain 24 h postoperative (3.0) and at discharge (2.0) [94]. In addition, a significant mean change was observed between the two groups in pain upon cough 6 h postop (LAP-TAP: 4.0, port site infiltration: 5.0) and on discharge (LAP-TAP: 1.5, port site infiltration: 3.0) [94]. On the contrary, the LAP-TAP block participants experienced higher pain scores (*p* = 0.043) and opioid requirements (*p* = 0.021) at 6 h than port infiltration group participants [84].

#### 15.3.12. Analgesic Efficacy of Laparoscopic-Assisted Infiltration Using Different Doses of Local Anesthetic versus Normal Saline

The results of this review show that patients who received 0.9% normal saline for infiltration during laparoscopy reported a high need for rescue analgesia both in the ward (21, 70%) and at PACU (13, 43.3%) [88]. Compared with the other two groups, a higher proportion of those who received Ropivacaine at low concentrations (0.25%) required rescue anesthesia in the ward (5 (16.7%) [88]. The values of the control group VAS were significantly higher than those of the groups TAI, TAPB, and IPLA after 1, 2, 4, 6, 12, and 24 h [90]. In addition, the VAS values of the IPLA group were significantly higher than those of the LAI and TAPB groups at 1, 2, 4, 6, 12, and 24 h [90]. VAS values at 1, 2, 4, 6, and 24 h were not significantly different between TAI and the TAPB group. A significant difference was observed between the TAI and TAPB groups in terms of VAS values at 12 h [90].

c.Other known techniques

When bilateral double transversus abdominis plane blockade (BD-TAP) was compared with sham control, higher pain sensation was observed postoperatively in the control group at rest (2 h), during coughing (2, 6, and 48 h), and during walking (2, 6, and 48 h), with no change between the two groups at 24 h [91]. Ultrasound-guided modified BRILMA block (US-BRILMA) was compared with US-TAP block (subcostal), and it was found that postoperative Morphine consumption was higher in the US-BRILMA group (5.67 mg) than in the US-TAP block group (5.17 mg) [93]. The time to request rescue analgesia was higher in the US-BRILMA group (845) than in the US-TAP group (759.33) [93]. The efficacy of erector spinae block and oblique subcostal transversus abdominis block was compared, and the mean pain of the two groups between 2 and 24 h differed, with the OSTAP block group reporting more pain experience (2.27 to 0.70) [92]). The US-TAPB block group had significantly lower resting VAS values than the two groups receiving PCIA with Sufentanil and Nalbuphineat 2, 6, 12, 24, and 48 h postop (*p* < 0.05) (Han et al., 2022). Postoperatively, dynamic VAS was significantly lower in the US-TAPB block group (*p* < 0.05) than in the two groups receiving PCIA with Sufentanil and Nalbuphine [98].

## 16. Long-Term Effects 

As shown in Table 3, all studies examined participants between 0 and 48 h or on the first postoperative day, with only one study examining pain intensity one week postop. Therefore, we cannot infer from this study that TAP block anesthesia has a long-term effect.

### Objective #4. Clinical Significance of TAP Block 

As shown in Table 4, the minimal clinically significant differences between the US-TAP block and port infiltration groups all had a large effect size index; however, the US-TAP block group had a higher effect.

## 17. Discussion

To our knowledge, this is the first review to comprehensively compare the analgesic efficacy of different procedures for laparoscopic cholecystectomy.

The seven classes of drugs used before anesthesia (pre-medication) include benzodiazepines (e.g., Midazolam), analgesics (e.g., Paracetamol), prokinetics (e.g., Metoclopramide), anticholinergics (Glycopyrrolate), 5-HT3 antagonists (Ondansetron), H2 receptor blockers (e.g., ranitidine), and alkalinizing agents (ringer lactate solution) [20,60,68,71,74,76]. Common local anesthetics used for various block procedures in laparoscopic cholecystectomy include Bupivacaine (0.25–0.375%), Ropivacaine (0.2–0.75%), and Levobupivacaine (0.25–0.375%). In all, 20 mL and 10 mL of local anesthetics could be the optimal dose for US-TAP or US-OSTAP blockade procedures and port infiltration. 

Evidence from this study has shown that US-TAP blockade in addition to general anesthesia is more effective than general anesthesia alone for postoperative pain management [59]. Use of the US-TAP block is associated with a reduced need for analgesics within the first 24 h postoperative [7]. In addition, using US-TAP block before induction of anesthesia and after the surgery might be associated with less need for analgesics within the first 24 h after laparoscopic cholecystectomy, although the two groups do not differ significantly [95]. The lower the concentration of local anesthetics used for laparoscopic infiltration, the higher the need for rescue analgesics [7]. Similarly, the higher the concentration of the solution used for US-TAP block, the lower the need for intra-operative or postoperative analgesia [7,72,98].

Apparently, the addition of 20 mL of normal saline to 0.5% bupivacaine for the US-TAP block might be associated with a lower need for analgesics than in patients who received 10 mL of normal saline to 0.5% bupivacaine [72]. US-TAP blockade with 0.375% Ropivacaine plus 2 mL Dexamethasone is more effective in relieving pain than US-TAP blockade with the same concentration of Ropivacaine or general anesthesia alone [67]. However, the addition of a lesser amount of Dexamethasone 1 mcg to 0.5% Ropivacaine does not result in any significant change [88]. Again, the same concentration of 0.2% Ropivacaine with an additional 4 mg/kg Dexamethasone was used for different TAP block procedures, and the US-ESP group had less postoperative pain and required fewer opioids than the OSTAP block group [92]. The addition of 0.5 mg/kg Esmolol to the local anesthetic for TAP block infiltration is more effective for pain management than the addition of 0.9% normal saline [89]. Again, administration of a local anesthetic percutaneously or subcutaneously or between obliquus internus and transversus abdominis or in the sub-diaphragmatic and pericholecystic areas is more effective than normal saline. In another development, the addition of 0.4% Ropivacaine to 100 mL of 0.9% saline and 10 mg/kg Tropisetron for US-TAP blockade could provide better pain relief than the addition of 2 mg/kg Sufentanil or Nalbuphine via PCA [98]. Patients receiving 0.5 mcg/kg Dexmedetomidine over 0.375% Ropivacaine consumed fewer opioids and took longer to request initial analgesia than patients receiving 2 mL of 0.9% saline over 0.375% Ropivacaine [75]. Magnesium Sulfate (0.5 g) was added on top of 0.25% Bupivacaine and achieved a reduction pain and in intra-operative opioid consumption [65]. The addition of 5 mcg Epinephrine to 0.375% Levobupivacaine is associated with lower postoperative and total opioid consumption [79]. Ketorolac (180 mg) was administered in combination with Oxycodone (40 mg) in addition to 100 mL of 0.9% saline, compared with patients receiving 20 mL of 0.2% Ropivacaine and those receiving additional patient-controlled analgesia; however, patients without local anesthetic had a greater need for analgesia than the others [71]. It was found that the addition of 2 mL of Sufentanil to 0.5% Bupivacaine prolonged the time to the first analgesia request in 24 h compared with patients who received 2 mL of normal saline or normal saline alone in addition to Bupivacaine [64].

US-TAP blockade with 2 mLof Sufentanil on 0.5% Bupivacaine might be associated with a lower opioids requirement than the use of 2 mL of 0.9% saline on 0.5% Bupivacaine or normal saline alone [64]. Uniquely, US-TAP blockade with 1% of Pethidine (40 mL) in addition to general anesthesia was found to be more effective for pain than US-TAP blockade with normal saline [70]. In particular, it is associated with lower intra-operative Fentanyl consumption and lower postoperative Morphine requirements [70].

Postoperative opioid consumption differs for the different types of blocks, with US-TAP consuming more Fentanyl and US-OSTAP consuming more Nalbuphine [61]. Although US-OSTAP was a relatively new technology, the results seemed consistent with US-TAP. Notably, when comparing the analgesic efficacy of the US-OSTAP and US-TAP block procedures, there appears to be no intra-operative difference between the two procedures in terms of Ketorolac (NSAID) consumption; however, US-OSTAP consumed more Fentanyl (opioids) than US-TAP block [61]. In contrast, more Fentanyl (opioids) was consumed postoperatively during the US-TAP block than during the US-OSTAP block [61]. However, Nalbuphine (opioids) consumption was higher in US-OSTAP block than in US-TAP block [61]. 

US-OSTAP blockade with Ropivacaine is more effective against pain and requires fewer analgesics than normal saline during and after surgery and at PACU [83]. In particular, US-OSTAP blockade appears to be associated with less pain and less need for opioids 2 h postop and at PACU compared with US-TAP [83]. US-TAP Subcostal block is associated with less pain experience and opioid consumption compared with quadratus lumborum [78]. Between 0 and 6 h postop, the US-TAP subcostal block appeared to be associated with higher pain perception; in contrast, the quadratus lumborum was perceived as more painful between 12 and 24 h [97]. To view it differently, quadratus lumborum reportedly had lower pain perception, lower cumulative daily Morphine consumption, and longer median time to first analgesic request compared with US-TAP block [97]. Therefore, further quantification is needed to clarify the pattern of analgesic consumption during different TAP blockade procedures.

Laparoscopically assisted TAP blockade with Bupivacaine plus Periportal injection of normal saline is associated with pain reduction between 1 and 6 h postoperative compared with those who received saline TAP blockade with a periportal injection of Bupivacaine [63]. The use of normal saline for LAP-TAP correlates with a higher need for rescue analgesia both in the ward and at PACU [88].

The port infiltration group seems to have a higher need for analgesics than the US-STA or US-TAP groups [94]. Notably, in the US-TAP block group, there seems to be a delay in requesting the first analgesics compared to the port infiltration group [94]. Within 3–6 h postoperative and at discharge, there appeared to be no significant difference between LAP-TAP and the port infiltration group; however, at 24 h, the port infiltration group had a greater demand for analgesia [94]. In contrast, the LAP-TAP group had a greater need for opioids and greater pain perception 6 h postoperative [94].

Subcostal US-TAP blockade is associated with lower postoperative opioid consumption compared with US-BRILMA [93]. Likewise, the time to request rescue analgesia was higher with the US-BRILMA blockade than with the US-TAP subcostal blockade [93]. In addition, US-ESP blockade was found to result in less pain between 2 and 24 h postoperative compared with OSTAP blockade [77]. Moreover, the bilateral double blockade of the transversus abdominis plane with Ropivacaine was associated with less pain perception at 2 h at rest and at 2, 6, and 48 h during walking and coughing, compared with the sham control group with normal saline [91].

The optimal dose means that symptoms and side effects can be most effectively controlled with the lowest dose of a drug [103]. From this study, we can infer that the optimal dose of local anesthetic for the blocks US-TAP or US-OSTAP could be 20 mL of Ropivacaine, Bupivacaine, or Levobupivacaine and 0.4 mL/kg for port site infiltration (Table 4). This is because we recorded a low number of side effects, adverse events, and complications (Table 4). However, caution should be exercised in using these data because of insufficient evidence. Overall, the effects of the interventions in this study were short-term effects, and we could not find evidence of long-term effects because in most cases, the studies assessed outcomes between 0 and 24 h, and very few studied subjects 48 h or a week postoperative.

Corroborating the extant literature, our study confirmed the reduction in analgesic consumption 24 h after laparoscopic surgery with TAP blockade or general anesthesia with TAP blockade compared with general anesthesia or no TAP blockade or placebo treatment, with analgesic consumption also reduced after 24 h [45]. Consistent with the results of our study, Kalu et al. [104] submitted that postoperative opioid consumption was influenced by the use of the US-TAP block procedure both preoperatively and postoperatively. Notably, there was no significant difference between groups in opioid consumption, but the US-TAP blockade reduced postoperative pain in both groups. From this review, the higher the concentration of local anesthetic used for local infiltration, the greater the effect on pain. Notably, this study found that 30 mL of 0.5% Levobupivacaine was more effective than 0.2% at 6 h postoperative; El-Dawlatly et al. [59] also used 30 mL of 0.5% Bupivacaine for US-TAP procedures and achieved a reduction in opioid consumption for 24 h postoperative. Similarly, 0.375% Ropivacaine was equally more effective against pain compared with 0.25% between 10 and 60 min after surgery when administered by the same route [69]. The route of local anesthetic administration has been controversial; however, in previous studies, different routes of administration of Ropivacaine were found to be more effective than Bupivacaine [105,106]. Although Bupivacaine and Ropivacaine have been compared at various concentrations in the context of different surgical procedures, there appears to be a paucity of evidence comparing these local anesthetics in US-TAP blockade for laparoscopic cholecystectomy. Therefore, future studies should compare the analgesic efficacy of Ropivacaine and Bupivacaine in different routes of administration for laparoscopic cholecystectomy.

The modified BRILMA block has been used and was found to reduce intra-operative Fentanyl consumption and postoperative Morphine consumption in supra-umbilical open surgeries such as cholecystectomy and gastrectomy [107]. OSTAP block was found to reduce postoperative pain scores more than intravenous multimodal analgesics, and TAP for laparoscopic cholecystectomy [68]. In a similar study, Basaran et al. [62] showed significant improvement in respiratory function and postoperative pain with OSTAP blockade. As our study shows, the OSTAP block reduced postoperative Tramadol consumption significantly more than the ESP block; however, the ESP block did not reduce postoperative Tramadol consumption as significantly as the OSTAP block.

Opioid consumption varies by plane block procedure. While there are no intra-operative differences between US-TAP and US-OSTAP for Ketorolac, the US-OSTAP group consumed more Fentanyl, while the US-TAP group consumed more Nalbuphine postoperatively using 0.375% of Ropivacaine 40 mL [61]. However, there is contradictory evidence in the literature. It is worth noting that the US-TAP block reduced intra-operative consumption of Remifentanil or Sufentanil when 30 mL of Bupivacaine or Levobupivacaine was used [7,59]. Ortiz et al. [20] used 30 mL of 0.5% Ropivacaine and achieved lower intra-operative consumption of Morphine and Fentanyl compared to the port site infiltration. In addition, analgesic consumption and the need for rescue analgesia were reduced [7,59]. To look at it another way, 20 mL of Ropivacaine was previously found to reduce pain when coughing but not at rest [21]. US-TAP blocks following general anesthesia were significantly associated with lower Morphine consumption in the 24 h following surgery compared with patients receiving general anesthesia alone [59]. Therefore, future studies should be designed to clarify the analgesic efficacy of different block procedures using similar dosages.

The results of this review have shown that the use of adjuvant in addition to local anesthetics for TAP block procedures could be effective for pain management in laparoscopic cholecystectomy. To improve recovery after surgery and reduce postoperative opioid consumption, opioid-sparing techniques are increasingly used in anesthesia. Evidently, it was found that local anesthetics may be improved, and additional analgesics need to be administered less frequently when Dexmedetomidine is added to local anesthetics during central neuraxial blocks and peripheral nerve blocks [108]. Additionally, a study has shown that postoperative Fentanyl requirements were significantly lower in patients in the Esmolol group [109]. Perineural Dexamethasone combined with posterior TAP block was found to have a prolonged analgesic effect [110]. The pharmacokinetics of Ropivacaine were studied after the addition of Epinephrine for abdominal trunk blocks; however, this was found to attenuate the systemic absorption of Ropivacaine [111]. Previous studies have shown that multimodal analgesia with TAP blockade in combination with Nalbuphine PCIA is likely to be more beneficial for hemodynamic stability than Sufentanil or Nalbuphine PCIA, which is in line with this study outcome [112]. In abdominal TAP procedures, Magnesium Sulfate in addition to Bupivacaine reduced opioid requirements, duration of anesthesia, and pain intensity without adverse effects [113]. For OSTAP blockade, Pethidine was used in comparison with Bupivacaine and normal saline, and it proved to be as effective as Bupivacaine. The result is consistent with previous studies on the efficacy of US-OSTAP blocks in laparoscopic cholecystectomy [47,60]. 

Study results by McDonnell et al. [114] and [115] suggest that local anesthetics in TAP are cleared only after 36 to 48 h, possibly because TAP has fewer blood vessels compared to other body regions. Because there are no blood vessels in TAP, there is less risk of systemic toxicity from local anesthetics, which can occur when blood vessels are punctured, a common complication of peripheral nerve blocks. An effective method for relieving abdominal pain is to block the abdominal wall nerves (intercostal nerves, T7-T12, and ilioinguinal and iliohypogastric nerves, L1) [7]. There are two nerves that cross the intercostal plane between the obliquus internus muscle and the transversus abdominis muscle [7]. TAP blocking eliminates the pain caused by abdominal distension due to pneumoperitoneum during four accesses to laparoscopic cholecystectomy, even though the gallbladder is a supra-umbilical organ [69]. Unlike conventional blind techniques, ultrasound allows direct visualization of the target plane, virtually eliminating the limitations of anatomic and marker access. In patients with limited cardiac status, TAP blockade has also been used as an effective analgesic during abdominal surgery [116,117]. 

A few studies have contributed to the understanding of the modulation of postoperative pain by Esmolol, although its role in modulating pain still remains unclear [89]. Analgesic effects of beta-adrenergic antagonists are mediated by G proteins that are activated in isolated cell membranes [108]. Clonidine, which also acts on G proteins, produces central analgesia by activating these proteins [108]. Clinical studies have shown that the use of Magnesium Sulfate as an adjunct to local anesthetics is effective for pain in regional procedures [7,21,59]. However, the mechanism by which it acts remains unclear. It has been postulated that it may potentiate analgesic effects through local or systemic actions [65]. For magnesium to exert analgesic effects, it must block calcium influx into nerve fibers and block the NMDA (*n*-methyl D-aspartate) receptors [118,119,120]. These effects may interfere with the release of neurotransmitters at synaptic junctions or enhance the effects of local anesthetics [121]. There are many sites in the body where this NMDA receptor is found, including nerve endings, and it plays a well-defined role in modulating pain and various other functions [122,123,124]; therefore, blocking NMDA receptors could prevent peripheral nociceptive stimuli from causing central sensitization [125]. Specifically, as magnesium prevents NMDA receptor activation, calcium and sodium influx into the cell and potassium outflow into space activate peripheral nociceptive stimulation, resulting in central sensitization and enhancement [65]. 

The mechanism by which Dexamethasone might affect pain management also remains unclear; however, the lack of local blood vessels makes the TAP blockade lasts a long time because of the slow breakdown of local analgesia [114,126]. Aside from that, the literature has shown that different approaches to TAP blockade affect nerves differently [38,127]. In the past, Dexamethasone prolonged intercostal nerve blockade in sheep when added to Bupivacaine microspheres [128]. In addition, blockade of the sciatic nerves was induced in rats with Dexamethasone microspheres added to Bupivacaine [129]. Increasing systemic absorption and intraneural clearance of local anesthetics may be decreased because the vasoconstrictive effects of Epinephrine antagonize their inherent vasodilator effects and they may be redistributed intraneurally [130]. With shorter-acting agents, Epinephrine significantly prolongs both infiltration anesthesia and peripheral nerve blockade; it may also increase blockade somewhat, but with Bupivacaine, it prolongs epidural or peripheral blockade only slightly [131]. Research has shown that the use of different analgesics for multiple targets can result in satisfactory postoperative pain management [98]. There are several previous studies indicating a reduction in postoperative pain scores and opioid consumption after classic mid-axillary blocks US-TAP; a sensory blockade occurs between dermatomes T6 and T10 when TAP subcostal is reached [132,133]. As a result, OSTAP is used in the upper abdomen to relieve pain [134].

## 18. Limitations 

The present systematic review has some limitations. Although all included studies were searched from different countries, we are subject to publication bias because this systematic review includes studies published in English only. Because of the lack of sufficient data, we could not draw conclusions about the clinical significance of the various TAP block procedures. We were also unable to provide information on the long-term effects of the TAP blockade procedures because of a lack of evidence. There were numerous RCTs whose data were not suitable for meta-analysis, either because of a pictorial representation of the data, different methods of measuring outcomes, or inappropriate statistical analysis (e.g., reporting median and mean values with a range) or the lack of baseline data. Furthermore, the ASA grade and BMI in the baseline data of patients could have an impact on the tolerance of local anesthetics. Again, the broader inclusion criteria and overall objective may be a limiting factor to consider when planning further studies. In view of these problems, it is advisable to not generalize conclusions from this study to broader clinical settings. However, further studies are needed to clarify the analgesic efficacy of different TAP block procedures at similar doses. The optimal long-term effect of local anesthetics in TAP blockade procedures and the toxicity of local anesthetics should be further investigated.

## 19. Conclusions

Four different types of anesthetics (Bupivacaine, Ropivacaine, Levobupivacaine, and Lidocaine) have been reported to be used in concentrations ranging from 0.2% to 0.375% for LC procedures. Ten different types of drugs (normal saline, Dexamethasone, Dexmedetomidine, Magnesium Sulfate, Oxycodone, Ketorolac, Epinephrine, Esmolol, Tropisetron, and Sufentanil) were reportedly used as supportive agents in addition to local anesthetics for LC. Although concentrations of LA may vary, 20 mL is probably the optimal dose for TAP block procedures and 0.4 mg/kg for port infiltration. However, further quantification is needed to clarify the optimal dose for different anesthetic concentrations. US-TAP blockades performed in addition to general anesthesia were more effective for pain than port infiltration or general anesthesia alone. Postoperative pain perception and opioid consumption were higher in those who received a US-TAP block after surgery than in those who received a block before induction; however, it took a shorter time for those who received a US-TAP block after surgery to require the first analgesics. US-TAP block with normal saline reportedly had higher opioid consumption in 24 h compared with those with Bupivacaine over normal saline or Bupivacaine over Sufentanil. It appears that the higher the concentration of the anesthetic used for US-TAP blockade, the lower the pain sensation, and that an adjuvant to LA could enhance its analgesic effect. Evidently, those who received Ropivacaine at low concentrations required rescue anesthesia in the ward. There seems to be no significant difference between the US-TAP and US-OSTAP. Time to first analgesic intake/request was higher in the groups with US-TAP block compared with quadratus lumborum. However, pain perception between 12 and 24 h was lower in the US-TAP group than in the quadratus lumborum group. This should be clarified in further studies. Compared with the port infiltrations, the LAP-TAP block group reportedly had less pain at rest in the first 3–6 h after surgery. The minimal clinically significant differences for both TAP block procedures and port infiltration appeared to have a large effect size index, but this should be taken with caution because of insufficient evidence. Subcostal US-TAP blockade may be correlated with lower postoperative opioid consumption and reduced need for rescue analgesics compared with US-BRILMA. US-ESP proved to be more effective than US-OSTAP block for postoperative pain within 24 h. Finally, multimodal analgesia could be another strategy for pain management. Analgesia with the TAP blockade significantly reduces opioid consumption and also provides effective analgesia.

## Figures and Tables

**Figure 1 jcm-11-06896-f001:**
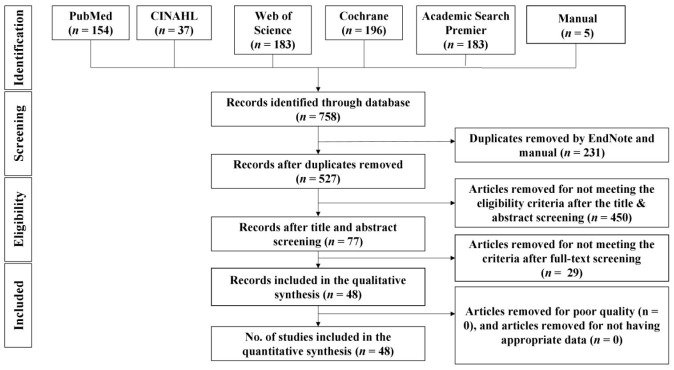
PRISMA flow chart.

**Table 1 jcm-11-06896-t001:** Quality appraisal using PEDro scale.

Author	Year	Eligibility	Randomized Allocation	Concealed Allocation	Similarity at Baseline	Blinding of Participants	Blinding of Therapist	Blinding of Assessor	Dropout	Intention to Treat	Group Comparison	PMVD	Total Score (10)	Internal Validity (8)	Sub Scale (2)	Interpretation
El-Dawlatly [59]	2009	Yes	Yes	Yes	Yes	No	Yes	No	Yes	Yes	Yes	Yes	8	6	2	Good
Ra [7]	2010	Yes	Yes	Yes	Yes	No	No	Yes	Yes	Yes	Yes	Yes	8	6	2	Good
Ortiz [20]	2012	Yes	Yes	Yes	Yes	Yes	No	Yes	Yes	No	Yes	Yes	8	6	2	Good
Petersen [21]	2012	Yes	Yes	Yes	Yes	Yes	Yes	No	Yes	Yes	Yes	Yes	9	7	2	Good
Tolchard [60]	2012	Yes	Yes	Yes	Yes	Yes	No	Yes	Yes	Yes	Yes	Yes	9	7	2	Good
Bhatia [6]	2014	Yes	Yes	Yes	Yes	No	No	Yes	Yes	Yes	Yes	Yes	8	6	2	Good
Shin [61]	2014	Yes	Yes	Yes	Yes	No	No	Yes	Yes	No	Yes	Yes	7	5	2	Good
Basaran [62]	2015	Yes	Yes	Yes	Yes	No	Yes	Yes	Yes	No	Yes	Yes	8	6	2	Good
Elamin [63]	2015	Yes	Yes	Yes	Yes	Yes	No	Yes	Yes	No	Yes	Yes	8	6	2	Good
Saliminia [64]	2015	Yes	Yes	Yes	Yes	Yes	No	Yes	Yes	Yes	Yes	Yes	9	7	2	Good
Al-refaey [65]	2016	Yes	Yes	Yes	Yes	No	No	No	Yes	Yes	Yes	Yes	7	5	2	Good
Bava [66]	2016	Yes	Yes	Yes	Yes	Yes	Yes	Yes	Yes	No	Yes	Yes	9	7	2	Good
Huang [67]	2016	Yes	Yes	No	Yes	Yes	Yes	Yes	Yes	No	Yes	Yes	8	6	2	Good
Oksar [68]	2016	Yes	Yes	No	Yes	No	Yes	No	Yes	No	Yes	Yes	6	4	2	Good
Sinha [69]	2016	Yes	Yes	Yes	Yes	No	Yes	Yes	Yes	No	Yes	Yes	8	6	2	Good
Breazu [70]	2017	Yes	Yes	No	Yes	No	No	Yes	Yes	No	Yes	Yes	6	4	2	Good
Choi [71]	2017	Yes	Yes	No	Yes	Yes	No	Yes	Yes	Yes	Yes	Yes	8	6	2	Good
Sahin [72]	2017	Yes	Yes	Yes	Yes	No	No	Yes	Yes	Yes	Yes	Yes	8	6	2	Good
Baral [73]	2018	Yes	Yes	No	Yes	No	No	No	Yes	Yes	Yes	Yes	6	4	2	Good
Bhalekar [74]	2018	Yes	Yes	Yes	Yes	Yes	No	Yes	Yes	Yes	Yes	Yes	9	7	2	Good
Sarvesh [75]	2018	Yes	Yes	No	Yes	No	No	No	Yes	No	Yes	Yes	5	3	2	Moderate
Suseela [76]	2018	Yes	Yes	No	Yes	Yes	No	Yes	Yes	Yes	Yes	Yes	8	6	2	Good
Altiparmak [77]	2019	Yes	Yes	Yes	Yes	Yes	No	Yes	Yes	No	Yes	Yes	8	6	2	Good
Baytar [78]	2019	Yes	Yes	Yes	Yes	Yes	No	Yes	Yes	No	Yes	Yes	8	6	2	Good
Houben [79]	2019	Yes	Yes	No	Yes	No	Yes	Yes	Yes	No	Yes	Yes	7	5	2	Good
Janjua [80]	2019	Yes	Yes	No	Yes	Yes	No	Yes	Yes	Yes	Yes	Yes	8	6	2	Good
Karnik [81]	2019	Yes	Yes	Yes	Yes	Yes	No	Yes	Yes	Yes	Yes	Yes	9	7	2	Good
Khandelwal [82]	2019	Yes	Yes	No	Yes	No	No	Yes	Yes	No	Yes	Yes	6	4	2	Good
Ribeiro [83]	2019	Yes	Yes	Yes	Yes	No	Yes	No	Yes	Yes	Yes	Yes	8	6	2	Good
Siriwardana [84]	2019	Yes	Yes	No	Yes	No	Yes	Yes	Yes	No	Yes	Yes	7	5	2	Good
Wu [85]	2019	Yes	Yes	Yes	Yes	No	No	Yes	Yes	Yes	Yes	Yes	8	6	2	Good
Arik [86]	2020	Yes	Yes	No	Yes	No	No	Yes	Yes	Yes	Yes	Yes	7	5	2	Good
Kharbuja [87]	2020	Yes	Yes	Yes	Yes	No	No	Yes	Yes	Yes	Yes	Yes	8	6	2	Good
Liang [88]	2020	Yes	Yes	Yes	Yes	No	Yes	Yes	Yes	No	Yes	Yes	8	6	2	Good
Abdelfatah [89]	2021	Yes	Yes	Yes	Yes	Yes	No	Yes	Yes	Yes	Yes	Yes	9	7	2	Good
Ergin [90]	2021	Yes	Yes	Yes	Yes	Yes	Yes	Yes	Yes	No	Yes	Yes	9	7	2	Good
Jung [91]	2021	Yes	Yes	Yes	Yes	Yes	Yes	Yes	Yes	Yes	Yes	Yes	10	8	2	Excellent
Sahu [92]	2021	Yes	Yes	Yes	Yes	No	Yes	Yes	Yes	No	Yes	Yes	8	6	2	Good
Saravanan [93]	2021	Yes	Yes	Yes	Yes	No	Yes	No	Yes	No	Yes	Yes	7	5	2	Good
Vindal [94]	2021	Yes	Yes	No	Yes	Yes	Yes	No	Yes	Yes	Yes	Yes	8	6	2	Good
Priyanka [95]	2022	Yes	Yes	No	Yes	No	No	No	Yes	No	Yes	Yes	4	2	2	Moderate
Emile [96]	2022	Yes	Yes	Yes	Yes	No	Yes	Yes	Yes	No	Yes	Yes	8	6	2	Good
Fargaly [97]	2022	Yes	Yes	Yes	Yes	Yes	No	Yes	Yes	Yes	Yes	Yes	9	7	2	Good
Han [98]	2022	Yes	Yes	No	Yes	No	No	No	Yes	Yes	Yes	Yes	6	4	2	Good
Lee [99]	2022	Yes	Yes	No	Yes	No	No	No	Yes	No	Yes	Yes	5	3	2	Moderate
Ozciftci [100]	2022	Yes	Yes	No	Yes	No	No	No	Yes	Yes	Yes	Yes	6	4	2	Good
Paudel [101]	2022	Yes	Yes	No	Yes	No	No	Yes	Yes	No	Yes	Yes	6	4	2	Good
Rahimzadeh [102]	2022	Yes	Yes	No	Yes	No	No	Yes	Yes	Yes	Yes	Yes	7	5	2	Good

PMVD = point measures and variability data. Note: Each item was scored either Yes = 1 or No = 0. Items 2−11 are summed for a PEDro total score. The sum of items 2−9 yields the internal validity subscale score, while the sum of items 10 and 11 yields the statistical reporting subscale score. The PEDro total score was rated 0−3 = poor, 4−5 = moderate, 6−8 = good, and 9−10 = excellent.

**Table 2 jcm-11-06896-t002:** Demographic and clinical characteristics of the participants.

No	Author (Year)	Sample Size	Gender, Age (Mean Age and/or Range, Ratio)	Pre-Medication	TAP Block Technique	Anesthetics Used for Surgical Infiltrations	Analgesia Used (Intra-Operative and Postoperative)	Use of PCA or PCIA	Outcomes	Outcome Measures
1	El-Dawlatly (2009) [59]	42	Gender (male *n* = 7, 16.7%; female = 35, 83.3%); Age TAP = 22–77 years; Control 34–65 years.	Lorazepam 2 mg, Ringers lactate 500 mL	US-TAP Block, bilateral Control (No TAP)	30 mL of Bupivacaine (5 mg/mL) 15 mL on each side (i.e., right and left).	**Intra-operative**: Sufentanil 0.1 mcg/kg **Postoperative**: Morphine 1.5 mg bolus, and total Morphine consumed in 24 h via PCIA were recorded.	Yes	Pain	NA
2	Ra (2010) [7]	54	Gender (male *n* = 28, 51.9%; female *n* = 26, 48.1%); Age: Control = 43.4 ± 12.4; US-TAP Block 0.25 = 48.2 ± 10.7; and US-TAP Block 0.5 = 45.0 ± 11.1.	None	US-TAP Block 1, bilateral US-TAP Block 2, bilateral Control (No TAP)	30 mL of Levobupivacaine 0.25%, 15 mL on each side (i.e., left and right). 30 mL of Levobupivacaine 0.5%, 15 mL on each side (i.e., left and right).	**Intra-operative**: Remifentanil **Postoperative**: Ketorolac 30 mg tds by 24 h, and Fentanyl 20 mcg for those with un-relived pain.	Yes	Pain	VNRS
3	Ortiz (2012) [20]	74	Gender (male *n* = 14, 18.9%; female *n* = 60, 81.0%); Age: US-STA Block = 37 (11); Control = 36 (11).	Midazolam 1–2 mg	US-TAP Block, bilateral Control (Port sites infiltration)	30 mL Ropivacaine 0.5%, 15 mL on each side (i.e., left and right). 20 mL to the port sites 7 mL for each of the 10 mm trocar sites, and 3 mL for each of the 5 mm trocar sites.	**Intra-operative**: Fentanyl 2 mcg/kg, additional 50 mcg bolus were added, and Morphine was given as needed at the end of the procedure. **Postoperative**: Ketorolac 30 mg.	No	Pain	NAS
4	Petersen (2012) [21]	74	Gender (male *n* = 53, 71.6%; female *n* = 21, 28.4%); Age: US-Posterior TAP (Ropivacaine) = 42 (13.5); US-Posterior TAP (saline) = 43 (17.0).	None	US-Posterior TAP Block, bilateral (Ropivacaine) US-Posterior TAP Block, bilateral (saline)	20 mL of Ropivacaine 0.5%, 10 mL on each side (i.e., left and right) + 2 mL saline. 20 mL of Normal saline, 10 mL on each side (i.e., left and right).	**Preoperative**: Remifentanil 0.4 mL/kg/h **Postoperative**: Acetaminophen 1000 mg by 4, Ibuprofen 400 mg by 3, Ketobemidone 2–24 h, and IV Morphine 0–2 h.	No	Pain	VAS
5	Tolchard (2012) [60]	43	Gender (male/female 2:0/5:16) Age: Intervention = 52 ± 3; Control = 48 ± 3.	Paracetamol 15–20 mg/kg; Diclofenac 0.5 mg/kg, Fentanyl 20 mcg	US-STA Block, bilateral, Control (Port site local infiltration)	Standardized dose of 1 mg/kg Bupivacaine	**Intra-operative**: Fentanyl 3 mcg/kg, Diclofenac 0.5 mg/kg, and Paracetamol 15–20 mg/kg. **Postoperative**: Fentanyl 20 mcg bolus.	No	Pain	VPAS
6	Bhatia (2014) [6]	64	Gender = NA; Age: Control = 35.4 ± 7.16; TAP Posterior = 36.4 ± 10.4; TAP Subcostal = 36.4 ± 10.4.	Alprazolam 0.25 mg, Ranitidine 150 mg	US-guided posterior TAP Block Subcostal US-TAP block. Control (Standard GA).	30 mL of Ropivacaine 0.375%, 15 mL on each side (i.e., left and right). 30 mL of Ropivacaine 0.375%, 15 mL on each side (i.e., left and right). General anesthesia only.	**Intra-operative**: Morphine 0.1 mg/kg **Postoperative**: Paracetamol 1000 mg every 6 h; IV Tramadol 2 mg/kg were given as an initial dose for those with VAS scores >4, with a subsequent dose of 1 mg/kg.	NA	Pain	VAS
7	Shin (2014) [61]	45	Gender (male *n* = 25, 53.2%; female *n* = 22, 46.8%); Age: Control = 44.7 ± 11.1; US-TAP group = 43.9 ± 9.5; and OSTAP group = 43.0 ± 9.6.	NA	US-OSTAP Block, bilateral US-TAP block, bilateral Control (GA only)	40 mL of 0.375% Ropivacaine. 20 mL on each side (i.e., left and right). 40 mL of 0.375% Ropivacaine. 20 mL on each side (i.e., left and right).	**Intra-operative**: Fentanyl 1 mcg/kg, Ketorolac 30 mg/kg (pre-emptive analgesia). **Postoperative**: Fentanyl 25 mcg for pain score >6, Ketorolac 30 mg for pain score 4–6, and Nalbuphine 10 mg for those needing analgesia at ward.	NA	Pain	VNRS
8	Basaran (2015) [62]	76	Gender (male *n* = 11, 14.5%; female *n* = 65, 85.5%); age: Control = 44.89 ± 14.2; Intervention = 43.2 ± 12.2.	Diazepam 10 mg	US-OSTAP Block, bilateral Control (GA only)	20 cc of 0.25% Bupivacaine on each side (i.e., left and right). General anesthesia only.	**Intra-operative**: Fentanyl 2 mcg/kg, 1 mcg/kg given (bolus) if heart rate or mean arterial pressure increased by 20% of initial values, Remifentanil 0.1 mcg/kg (maintenance), and 0.5 mg/kg Meperidine prior to the cessation of Remifentanil. Ephedrine 5 mg was given to reduce mean arterial pressure with an additional dose permitted after 2 min. IV Tenoxicam 20 mg after induction. **Postoperative**: Tramadol 50 mg IV on request with minimum of 20 min between doses, and a maximum dose was capped at 500 mg at 24 h.	NA	Pain	VAS
9	Elamin (2015) [63]	80	Gender (male *n* = 10, 12.5%, female 70, 87.5%); Age = 49.5 years versus 52.1 years.	None	LAP-TAP Block (Bupivacaine), bilateral, subcostal plus (Periportal saline injection) LAP-TAP (saline), bilateral, subcostal plus (Periportal Bupivacaine injection)	TAP (50 mL of 0.25% Bupivacaine), Periportal (20 mL Normal saline), and intraperitoneal (10 mL of 0.25%) Bupivacaine. 10 mL each to anterior axillary and mid-clavicular lines; bilateral infiltration in the petite triangle 15 mL each. TAP (50 mL of Normal saline), Periportal (20 mL of 0.25% Bupivacaine), intraperitoneal (10 mL of 0.25% Bupivacaine).	**Intra-operative**: NA **Postoperative**: Paracetamol 1 g q6h, Diclofenac sodium 75 mg.	Yes	Pain	NRS
10	Saliminia (2015) [64]	54	Gender (male *n* = 24, 24.4%; female *n* = 30, 54.6%); Age = 28–61 years.	None	US-TAP Block, bilateral, Bupivacaine + normal saline US-TAP Block, bilateral, (Bupivacaine + Sufentanil) US-TAP Block, bilateral, (normal saline only)	32 mL (Bupivacaine 30 mL + 2 mL of Sufentanil). 16 mL on each side (i.e., left and right). 30 mL of Bupivacaine + 2 mL of normal saline. 16 mL on each side (i.e., left and right). 32 mL of Normal saline. 16 mL of 0.9% Normal saline on each side.	**Intra-operative**: Fentanyl 3 mcg/kg, with 1 mcg/kg as maintenance dose. **Postoperative**: 50 mL of Fentanyl bolus with a lockout time of 8 min.	Yes	Pain	VAS
11	Al-refaey (2016) [65]	90	Gender (NA); Years = Control = 32 ± 6; US-TAP Block B= 37 ± 8; US-TAP Block M = 34 ± 8.	None	US-TAP Block, bilateral subcostal. US-TAP Block, bilateral subcostal GA only.	20 mL Bupivacaine 0.25%. 20 mL of Bupivacaine 0.25% + 0.5 g of magnesium sulphate. Anesthesia only.	**Intra-operative**: Fentanyl 1 mcg/kg **Postoperative**: Morphine 0.02 mg/kg bolus	No	Pain	VAS
12	Bava (2016) [66]	42	Gender (male *n* = 3, 7.1%; female *n* = 39, 92.3%); Age: TAP group = 33.7 ± 10.5, Control = 33.5 ± 6.5.	None	US-TAP Block, bilateral Control (port site infiltration)	30 mL 0.365% Ropivacaine. 15 mL on each side (i.e., left and right). 10 mL of 0.25% Bupivacaine.	**Intra-operative**: Fentanyl 2 mcg/kg and 0.5 mcg was used as supplemental. **Postoperative**: Morphine 0.5 mg/kg with a maximum dose of 20 mg in 4 h.	Yes	Pain	VAS
13	Huang (2016) [67]	60	Gender: NA Age: Control 1: 38.5 ± 7.7; Group II: 39.7 ± 5.5; Group III: 38.6 ± 8.9.	None	General anesthesia; US-TAP Block, bilateral; US-TAP Block, bilateral + 2 mLDexamethasone	GA only 30 mL of 0.375% Ropivacaine 7.5 mL/kg. 15 mL on each side (i.e., left and right). 32 mL (30 mL of 0.375% Ropivacaine + 2 mL Dexamethasone. 16 mL on each side (i.e., left and right).	**Intra-operative**: Remifentanil until its plasma concentration reaches 2.5 mcg/mL. **Postoperative**: Sufentanil 5–10 mcg.	No	Pain	NRS
14	Oksar (2016) [68]	60	Gender (male = 17, 28.3%; female 43, 71.7%); Age: 18–74.	Midazolam 2 mg IV, Ringers’ lactate solution 500 mL	Intercostal-iliac US-TAP block, bilateral + PCA, US-OSTAP + PCA, bilateral; GA + PCA alone	40 mL Lidocaine (5 mg/mL). 20 each to the left and right 40 mL Lidocaine (5 mg/mL). 20 each to the left and right.	**Intra-operative**: Remifentanil **Postoperative**: Paracetamol 1 g, and Diclofenac 75 mg. Pain relief using PCA was by 200 mg Tramadol (7 mL, 2 mg/kg bolus) with a 15 min lockout time.	Yes	Pain	VAS
15	Sinha (2016) [69]	60	Gender: (NA); Age: >40 years.	Oral Ranitidine 150 mg and alprazolam 0.25 mg	US-TAP block (Bupivacaine), bilateral US-TAP block (Ropivacaine), bilateral	40 mL of 0.25% Bupivacaine. 20 mL each to the right and left. 40 mL of 0.375% Ropivacaine. 20 mL each to the right and left.	**Intra-operative**: Fentanyl 2 mcg/kg. **Postoperative**: Diclofenac sodium 75 mg.	No	Pain	VAS
16	Breazu (2017) [70]	74	Gender (male 29, 39.2%; female 45, 60.8%); Age: 42–65 years OSTAP-placebo; 38–67 years OSTAP-Bupivacaine; 40–65 OSTAP-Pethidine.	7.5 mg Midazolam	US-OSTAP-placebo, bilateral; OSTAP-Bupivacaine, bilateral; OSTAP-Pethidine, bilateral.	40 mL of sterile saline (20 mL on each side). 40 mL of 0.25% Bupivacaine. 20 mL for each of the right and left. 20 mL of 1% Pethidine 10 mL on each side.	**Intra-operative**: Fentanyl 2 mcg/kg **Postoperative**: Pethidine 25–50 mg. at the ward level, Acetaminophen 1 g 8-hourly; however, those with moderate to severe pain continue to receive 25–50 mg of Pethidine until the VAS score is lower than 3.	Yes	Pain	VAS
17	Choi (2017) [71]	103	Gender: (male *n* = 48, 46.6%; female *n* = 55, 53.4%); Age: IV-PCA + GA (Control): 50.4 ± 15.9; US-TAP block: 49.1 ± 14.2; TAP block: 52.2 ± 11.8.	Midazolam 0.05 mg/kg, Glycopyrrolate 0.003 mg/kg	PCA + GA US-TAP block (indwelling catheter) US-TAP block+ PCA	100 mL of Normal saline + 40 mg Oxycodone and 180 mg of Ketorolac via IV-PCA pump. 20 mL of 0.2% Ropivacaine 20 mL of 0.2% Ropivacaine.	**Intra-operative**: Remifentanil 1 mcg/kg and 0.5–1 mcg was used for maintenance. **Postoperative**: Morphine 3–5 mg was given for unrelieved pain.	Yes	Pain	NRS
18	Sahin (2017) [72]	60	Gender: (male *n* = 33, 55%; female *n* = 27, 45%); Age: Group 1: 47.2 ± 13.0; Group 2: 64.5 ± 11.5.	No	Group 1. US-TAP block, unilateral (right sided) Group 2. US-TAP block, unilateral	30 mL (20 mL: 50 mg of Bupivacaine 0.5% + 10 mL of normal saline). 30 mL of 50 mg Bupivacaine plus 20 mL of normal saline.	**Intra-operative**: Fentanyl 2 mcg/kg. **Postoperative**: Diclofenac 25 mg when the VAS is < 7.	No	Pain	VAS
19	Baral (2018) [73]	60	Gender: (male *n* = 19, 31.7%; female *n* = 41, 68.3%); Age Subcostal TAP block 42.47 ± 14.41; Control: 45.93 ± 14.34.	No	US-TAP block, Subcostal Control (Port site infiltration).	20 mL of Bupivacaine. 10 mL on each side (i.e., left and right). 20 mL of 0.25% Bupivacaine. 5 mL to each of the 4 port sites).	**Intra-operative**: Fentanyl 2 mcg/kg. **Postoperative**: Pethidine 0.5 mg/kg if the VAS score is less than equal to 4.	No	Yes	VAS
20	Bhalekar (2018) [74]	50	Gender: US-TAP (saline): (male = 11(44); female = 14(56.00); US-TAP block: male 14(56.00); female 11(44.00). Age: Subcostal TAP block = 44.1 ± 13.1; Control: 44.1 ± 13.3.	0.2 mg glycopyrrolate, Ranitidine 50 mg and Ondansetron 4 mg.	US-TAP block, Subcostal, bilateral US-TAP block (saline)	40 mL of Bupivacaine 0.25%. 20 mL on each side (i.e., left and right). 40 mL of 0.9% normal saline. 20 mL on each side (i.e., left and right).	**Intra-operative**: Fentanyl 2 mcg/kg; Diclofenac 75 mg administered after induction. **Postoperative**: Nalbuphine 10 mg/70 kg with a further dose of 5 mg/kg when required.	No	Pain	VAS
21	Sarvesh (2018) [75]	60	Gender: (NA); Age > 50 years.	Midazolam 0.03 mg/kg,	US-TAP Block 1, Subcostal, bilateral, US-TAP Block 2, Subcostal, bilateral,	18 mL of 0.375% Ropivacaine+ 2 mL of normal saline. 20 mL on each side (i.e., left and right). 18 mL. 0.375% Ropivacaine with 2 mL of 0.5 μg/kg Dexmedetomidine 2 mL. 20 mL on each side (i.e., left and right).	**Intra-operative**: Fentanyl 2 mcg/kg **Postoperative**: Morphine 1 mg loading dose with a lockout time of 10 min, and 0.25 mg/kg 4 h limit.	Yes	Pain	NRS
22	Suseela (2018) [76]	80	Gender: (NA); Age: US-TAP Block = 42.25 ± 11.91; Control (Port site infiltration) = 41.00 ± 11.34.	Metoclopramide 10 mg and Ranitidine 150 mg and midazolam 0.5 mg.	US-TAP Block, Subcostal, bilateral, Control (Port site infiltration).	40 mL of 0.25% Bupivacaine. 20 mL on each side (i.e., left and right). 20 mL (0.5% Bupivacaine 5 mL each at 4 ports). 5 mL to each of the 4 ports.	**Intra-operative**: Fentanyl 2 mcg/kg and Paracetamol 1 g. **Postoperative**: Paracetamol 1 g 8-hourly, Tramadol 1 mg/kg bolus, Diclofenac 1 mg/kg.	No	Pain	NRS
23	Altiparmak (2019) [77]	68	Gender: (male 25, 36.8%; female *n* = 43, 63.2%); Age: US-OSTAP Block = 53.1 ± 14.7; US-ESP Block = 51.1 ± 12.3.	No	US-OSTAP Block, bilateral US-ESP Block, bilateral	40 mL (0.375% Bupivacaine). 20 mL on each side (i.e., left and right). 40 mL (0.375% Bupivacaine). 20 mL on each side (i.e., left and right).	**Intra-operative**: Fentanyl 1 mcg/kg **Postoperative**: Trometamol 50 mg, Tramadol 10 mg bolus with 20 min lockout time.	Yes	Pain	NRS
24	Baytar (2019) [78]	107	Gender: (male *n* = 26, 24.3%; female *n* = 81, 75.7%); Age QL Block: 46.42 ±16.57; US-TAP Block: 48.12 ± 12.42.	Midazolam 0.01–0.02 mg/kg	US-TAP Block, subcostal, bilateral quadratus lumborum block, bilateral	40 mL of 0.25% Bupivacaine. 20 mL for each side. 40 mL of 0.25% Bupivacaine. 20 mL for each side.	**Intra-operative**: Fentanyl 1–2 mcg **Postoperative**: Tenoxicam 20 mg, 54 mL normal saline + Tramadol 300 mg (6 mL).	Yes	Pain	VAS
25	Houben (2019) [79]	52	Gender: male *n* = 17, 32.7%; female *n* = 35, 67.3%; Age: US-TAP Block = 50.6 ± 12.9; Control (saline) = 47.5 ± 16.0.	Oral Etoricoxib 120 mg	US-TAP Block, subcostal, bilateral (Levobupivacaine) US-TAP Block, subcostal, bilateral, (saline)	40 mL Levobupivacaine 0.375% + Epinephrine 5 mcg/mL. 20 mL for each side. 40 mL 0.9% saline + Epinephrine 5 mcg/mL. 20 mL for each side.	**Intra-operative**: Sufentanil 0.1 mcg/kg **Postoperative**: Ketamine, Paracetamol 2 g (1 g for those with weight < 60 kg, and Morphine 2 mg bolus.	No	Pain	VAS
26	Janjua (2019) [80]	100	Gender: (male-female ratio = US-TAP Block 1.8: 2.6; Control (Port Site Infiltration): 1.7:2.8); Age: US-TAP Block = 48.70 ± 12.25; Port Site Infiltration = 48.35 ± 13.89.	No	US-TAP Block, unilateral Control (Port Site Infiltration)	0.25% Bupivacaine 0.4 mL/kg (1/3 to the fascial plane). 0.25% Bupivacaine 0.4 mL/kg, 1/3 intraperitoneally before the closure of the port sites.	**Intra-operative**: Nalbuphine 0.15 mg/kg, and Ketorolac 0.45 mg/kg **Postoperative**: Ketorolac 0.45 mg/kg by 2 8-hourly.	No	Pain	VAS
27	Karnik (2019) [81]	80	Gender: (male = 63, 78.8%; female 17, 21.2%); Age: US-TAP Block = 6.3 ± 3.8; Local infiltration = 5.5 ± 2.9.	Midazolam 0.05 mg/kg	US-TAP Block., bilateral Control (port sites local infiltration)	40 mL of 0.25% Bupivacaine. 20 mL for each side. 0.25% Bupivacaine. 0.4 mL/kg.	**Intra-operative**: Fentanyl 2 mcg/kg, 1 mcg/kg as maintenance, and Paracetamol 15 mg/kg **Postoperative**: Diclofenac 1 mg/kg.	No	Pain	VAS
28	Khandelwal (2019) [82]	80	Gender (male = 25, 31.25%; female = 55, 68.75%); Age: US-STA Block = 42 ± 9.4; Control (intraperitoneal infiltration) = 44 ± 8.6.	No	US-STA Block, subcostal, bilateral Control (intraperitoneal infiltration)	40 mL of 0.25% Levobupivacaine. 20 mL on each side (i.e., left and right). 40 mL of 0.25% Levobupivacaine diluted with normal saline. 40 mL intraperitoneally.	**Intra-operative**: Fentanyl 2 mcg/kg. **Postoperative**: Tramadol 1 mg/kg.	No	Pain	NRS
29	Ribeiro (2019) [83]	42	Gender: (male = 27, 64.3%; female = 15, 35.7%); Age: US-OSTAP Block (Ropivacaine) = 45.45 ± 14.12; US-OSTAP Block (Normal saline) = 40.05 ± 11.91.	No	US-OSTAP Block (Ropivacaine), bilateral US-OSTAP Block (Normal saline), bilateral	40 mL of 0.35% Ropivacaine. 20 mL on each side (i.e., left and right). 40 mL of sterile normal saline. 20 mL on each side (i.e., left and right).	**Intra-operative**: Paracetamol 1 g **Postoperative**: Paracetamol 1 g 8-hourly, and Tramadol 1 mg/kg when pain threshold exceeds 4.	No	Pain	VAS
30	Siriwardana (2019) [84]	90	Gender: male-female ratio LAP-TAP = 0.214; Control = 0.333; (females: 72.2%; Age: 19–80 years).	No	LAP-TAP Block, subcostal + Port Site Infiltration Control (Port Site Infiltration).	40 mL of 0.25% Bupivacaine 20 mL on each side (i.e., left and right) + 3–5 mL of standard port site infiltration. 3–5 mL of 0.25% Bupivacaine standard port site infiltration.	**Postoperative**: Morphine 0.1 mg/kg.	Yes	Pain	Unspecified
31	Wu (2019) [85]	160	Gender: (male = 124, 77.5%; female 56, 22.5%); Age: LA1 = 48.0 ± 11.4; TL = 47.6 ± 10.1; TR = 48.6 ± 12.1.	No	LAI –Group TL-Group (US-TAPB + LAI) TR-Group (US-TAPB + RSB)	30 mL of 0.5% of Ropivacaine + 1 mcg/kg of Dexamethasone. 30 mL of 0.25% Ropivacaine (i.e., 15 mL left and right). + 1 mcg of Dexamethasone. Pre-incisional infiltration: 30 mL 0.25% of Ropivacaine + 1 mcg of Dexamethasone. Plus 40 mL (20 mL to the left and right each), and 20 mL to the bilateral rectus sheath.	**Intra-operative**: Flurbiprofen Axetil 1.5 mg/kg, and Remifentanil 1 mcg/kg. **Postoperative**: Flurbiprofen Axetil 1.5 mg/kg 6-hourly.	No	Pain	VAS
32	Arik (2020) [86]	72	Gender: (Male = 16, 23.6%; female = 56, 76.4%); Age: TAP Block = 42.8 ± 9.2; Local Anesthetic infiltration = 42.9 ± 11.2; IV-PCA = 46.6 ± 13.8.	No	TAP Block, Unilateral Subcostal Port site local infiltration IV-PCA only	22 mL (0.25% Bupivacaine, and 2 mL saline). 20 mL of 0.25% of Bupivacaine.	**Intra-operative**: Remifentanil infusion **Postoperative**: Tramadol 5 mg/mL, 20 mg bolus with 20 min lockout time with a maximum of 200 mg per 4 h.	Yes	Pain	NRS
33	Kharbuja (2020) [87]	60	Gender: (male = 16, 26.7%; female 44, 73.3%); Age: Subcostal TAP = 40.27 ± 12.57; Control (Port Site Infiltration) = 38.77 ± 9.95.	Ranitidine 150 mg.	US-TAP Block, subcostal, bilateral Control (Port Site Infiltration)	40 mL of Bupivacaine 0.25% 20 mL to each side. 20 mL of 0.5% Bupivacaine 5 mL at each port.	**Intra-operative**: Fentanyl 2 mcg/kg and Paracetamol 1 g. **Postoperative**: Fentanyl 20 mcg/kg, and Paracetamol 1 g 8-hourly.	No	Pain	VAS
34	Liang (2020) [88]	120	Gender: (male 43, 35.8%; female 77, 64.2%); Age: Group H = 49.5 ± 12.1; Group M 50.0 ± 13.0; Group L = 47.2 ± 13.9; Group C = 51.5 ± 12.8.	No	Group H wound infiltration port Group M wound infiltration port Group L wound infiltration port Group C (Control)	20 mL of 0.75% Ropivacaine 20 mL of 0.5% Ropivacaine 20 mL of 0.2% Ropivacaine. 20 mL of 0.9% Normal saline.	**Intra-operative**: Fentanyl 3 mcg/Kg), and maintenance using Remifentanil, at a dose of 0.1 mg/kg/hour. **Postoperative**: Parecoxib 40 mg, Morphine 2.5 mg (rescue) for those at PACU, and 100 mg (rescue) for those at ward.	No	Pain	NRS
35	Abdelfatah (2021) [89]	60	Gender: (female 51, 85%; male 9, 15%); Age: US-TAP Block 1 = 32.66 ± 10; US-TAP Block 2 = 31.67 ± 10.7.	No	US-TAP Block 1 (0.25% Bupivacaine + Esmolol) US-TAP Block 2 (0.25% Bupivacaine + Isotonic Saline)	40 mL Bupivacaine (i.e., 20 mL on each side) + Esmolol 0.5 mg/kg. 40 mL Bupivacaine + 30 mL isotonic saline (loading dose), and 0.05 mg/kg/min (maintenance dose).	**Intra-operative**: Fentanyl 1–2 mcg/kg, **Postoperative**: Fixed dose of Acetaminophen 500 mg/6 h, Morphine 5 mg.	No	Pain	VAS
36	Ergin (2021) [90]	160	Gender: (male 41, 25.62%; female 119, 74.38%); Age = 18–74 years.	No	TAI-Group (administered percutaneously and subcutaneously) TAPB-Group (solutions administered to the left in between two muscles i.e., internal oblique and transversus abdominis. IPLA-Group (administer to the sub-diaphragmatic and pericholecystic areas). Control (no local anesthetic)	20 cc of 0.5% Bupivacaine + 20 cc of physiological saline. 20 cc of 0.5% Bupivacaine solution. 10 cc on each side (i.e., left and right). 20 cc of 0.5% Bupivacaine. No local anesthetics	**Intra-operative**: Paracetamol 1 g **Postoperative**: Tramadol 50 mg, and 100 mg for those with ongoing pain, and tabs Tenoxicam 20 mg 8-hourly.	No	Pain	VAS
37	Jung (2021) [91]	76	Gender: (male = 32, 42.1%; female 44, 57.9%); Age: BD-TAP = 48.9 ± 8.3; Control 47.5 ± 8.7.	No	BD-TAP Block, bilateral Control (Sham Block), bilateral	60 mL >50 kg, 15 mL of 0.25% Ropivacaine; < 50 kg more diluted 3 mg/kg. 30 mL on each side (i.e., left and right). 60 mL of 0.9% Normal Saline. 30 mL on each side (i.e., left and right).	**Intra-operative**: Remifentanil 2–6 μg/mL, Paracetamol 1 g, and Ibuprofen 400 mg. **Postoperative**: Oxycodone 3 mg (rescue), Ketorolac 30 mg (Day 0–1), and Tramadol 50 mg 8-hourly (from Day 1).	No	Pain	NRS
38	Sahu (2021) [92]	60	Gender: (male 35, 58.3%; female 25, 41.7%); Age: US-ESP Block 41.3 ± 11.8; OSTAP Block: 40.2 ± 11.1.	Midazolam 1 mg, Glycopyrrolate 0.2 mg	US-ESP Block, Bilateral OSTAP Block, Bilateral	40 mL of 0.2% Ropivacaine + 4 mg of Dexamethasone. 20 mL each to the left and right. 40 mL of 0.2% Ropivacaine + 4 mg of Dexamethasone. 20 mL on each side (i.e., 20 mL left and right).	**Intra-operative**: Nalbuphine 0.1 mg/kg. **Postoperative**: Paracetamol 1 g 4-hourly x 24 h, Tramadol 1 mg/kg (rescue), and when pain persists, Diclofenac 75 mg was used as second option.	No	Pain	VAS
39	Saravanan (2021) [93]	60	Gender: (male = 26, 43.3%, female 34, 56.7%); Age: US-Modified BRILMA Block = 47.7 ± 11.12; Subcostal TAP Block 42.8 ± 11.09.	No	US-Modified BRILMA Block, US-TAP Block, subcostal	20 mL 0.2% Ropivacaine 20 mL 0.2% Ropivacaine.	**Intra-operative**: Fentanyl 2 μg/kg, with 1 mcg/kg as maintenance dose, and Paracetamol 1 g. **Postoperative**: Morphine 0.1 mg/hour with a bolus of 1 mg, and lockout time of 10 min.	Yes	Pain	VAS
40	Vindal (2021) [94]	100	Gender: (male = 11, 11%; female = 89, 89%); Age: TAP Block 35(15.5); Port Site Infiltration: 35(18.25).	No	LAP-TAP Block Control (Port Site Infiltration)	40 mL 0.25% Bupivacaine. 10 mL at each of the four marked sites. 40 mL 0.9% Normal Saline. 10 mL at each of the 4 port sites.	**Intra-operative**: NA **Postoperative**: Diclofenac sodium 50 mg (rescue) and 50 mg when needed.	No	Pain	VAS
41	Priyanka (2022) [95]	80	Gender: (male = 23, 33.3%; female 46, 66.7%); Age: US-TAP Block pre: 45.40; US-TAP Block post: 45.29.	The night before surgery: Ranitidine 150 mg, and Tabs Alprazolam 0.5 mg Prior to surgery: Glycopyrrolate 0.005 mg/kg, Midazolam 0.05 mg/kg, and Fentanyl 2 mcg/kg.	US-TAP Block Pre, bilateral US-TAP Block Post, bilateral	40 mL of 0.25% Bupivacaine. 20 mL spread to the left and right. 20 mL of 0.25% Bupivacaine. 20 mL spread to the left and right.	**Intra-operative**: Fentanyl 2 mcg/kg **Postoperative**: Tramadol 100 mg	No	Pain	VAS
42	Emile (2022) [96]	110	Gender: (male 11, 10%, female 99, 90%); Age: 40.9 ± 11.7.	No	US-TAP Block, bilateral LSTAP Block Control (GA only)	20 mL of 0.25% of Bupivacaine + 2% Lidocaine (i.e., 10 mL left and right) + normal saline. 20 mL of 0.25% of Bupivacaine + 2% Lidocaine + normal saline. (i.e., 10 mL left and right).	**Intra-operative**: NA **Postoperative**: Paracetamol 1000 mg and Diclofenac were used for unsatisfactory pain relief.	No	Pain	VAS
43	Fargaly (2022) [97]	50	Gender: (male = 8, 16%; female 42, 84%); Age: US-TAP Block = 33.2 ± 9.1; QL Block = 32.7 ± 8.4.	No	US-TAP Block Group, bilateral QLB-Group, bilateral	40 mL of 0.25% Bupivacaine. 20 mL for each side (i.e., right and left) 40 mL of 0.25% Bupivacaine. 20 mL for each side (i.e., right and left).	**Intra-operative**: Fentanyl 1μgkg. **Postoperative**: Paracetamol 1 g 8-hourly, and Ketorolac30 mg 12-hourly. Morphine sulfate 3 mg bolus increments with the highest amount of 15 mg/4 h or 45 mg a day.	No	Pain	VAS
44	Han (2022) [98]	180	Gender: (male = 124, 68.9%; female = 56, 31.2%); Age: Group S = 45.78 ± 17.13; Group N = 44.52 ± 17.71; US-TAPB Block = 46.28 ± 13.18.	No	US-TAPB Block Group Group S Group N	20 mL Ropivacaine 0.4% + 10 mg of Tropisetron diluted with normal saline 0.9% 100 mL. 40 mL of 0.25% Bupivacaine + 1 mL saline. 20 mL + 1 mL (for each of the blocks) Sufentanil 2 mg/kg via PCIA pump, + 10 mg of Tropisetron diluted with normal saline 0.9% 100 mL. Nalbuphine 2 mg/kg via PCIA pump, + 10 mg of Tropisetron diluted with normal saline 0.9% 100 mL.	**Intra-operative**: Sufentanil 0.4–0.6 mcg/kg, Remifentanil 0.05–0.2 mcg/h.	Yes	Pain	VAS
45	Lee (2022) [99]	53	Gender: (male = 31, 54.5%; female = 22, 44.5%); Age: 1. US-TAPB –Block = 44.3 ± 9.8; Control = 45.7 ± 12.0.	No	US-TAP Block (0.375% Ropivacaine), US-TAP Block (Normal saline).	40 mL of 0.375% Ropivacaine. 20 mL per side (i.e., right and left). 40 mL of 0.9% Normal Saline. 20 mL per side (i.e., right and left).	**Intra-operative**: Remifentanil 0.5 mcg/kg and 0.1 mcg/kg/min as maintenance dose. **Postoperative**: Fentanyl 0.2 mcg/kg bolus and every hour with a 15 min lockout time.	Yes	Pain	VAS
46	Ozciftci (2022) [100]	90	Gender: (male = 24, 26.7%; female 66, 73.3%); Age: Control: 47.46 ± 11.83; TAP Block, unilateral 48.46 ± 12.05; TAP Block, bilateral: 51.90 ± 11.40.	Midazolam 0.02 mg/kg	TAP Block, Unilateral (right side). TAP Block, Bilateral Control	20 mL of 0.25% Bupivacaine (right only). 40 mL of 0.25% Bupivacaine. 20 mL for each of side.	**Intra-operative**: Paracetamol 1 g, Tramadol 2 mg/kg, Diclofenac sodium 75 mg. **Postoperative**: Paracetamol 1 g, Diclofenac sodium, and Tramadol 0.5 mg/kg hourly to a maximum of dose of 500 mg/day.	Yes	Pain	VNRS
47	Paudel (2022) [101]	60	Gender: (male = 14, 23.3%; female = 46, 76.7%); Age: TAP-Block: 41.63 ± 11.99; Control (local infiltration): 40.23 ± 11.42.	Ranitidine 150 mg	US-TAP Block, subcostal, bilateral Control (Local Port Sites Infiltration).	40 mL of 0.25% Bupivacaine. 20 mL to each side (i.e., right and left). 20 mL of 0.25% Bupivacaine at the port sites.	**Intra-operative**: Fentanyl 2 mcg/kg. **Postoperative**: NA	No	Pain	VAS
48	Rahimzadeh (2022) [102]	76	Gender: NR; Age: US-TAP (post-surgery) Block = 44.46 ± 8.30; US-TAP (after induction of anesthesia) = 45.0 ± 10.87.	Fentanyl 2 mcg/kg and Midazolam 0.12 mg/kg	US-TAP Block, bilateral (postop group) Pre-emptive Group, block (after induction)	40 mL of 0.25% Ropivacaine. 20 mL to the right and left. 40 mL of 0.25% Ropivacaine. 20 mL to the right and left.	**Intra-operative**: Fentanyl 2 μg/kg **Postoperative**: Acetaminophen20 mg/mL, and Ketorolac0.6 mg/mL bolus and 2 mL every 15 min.	Yes	Pain	NRS
	Total study population: N = 3651; Male = 1090, 29.9%; Female = 1822, 49.9%; Unspecified genders = 739, 20.2%. Measures: VAS = 30, 62.5%; NRS = 12, 25%; VNRS = 3, 6.25%; Unspecified = 1, 2.1%; VAPA = 1, 2.1%; NA = 1, 2.1%.									

GA = general anesthesia; VNRS = Verbal Numerical Rating Scale; PCA = patient-controlled analgesia; IV-PCA = intravenous patient-controlled analgesia; PCIA = patient-controlled intravenous analgesia; NA = not available; NAS = Numeric Analogue Scale; VAS = Visual Analog Scale; NR = not reported; NRS = Numerical Rating Scale; US-TAP = ultrasound-guided transversus abdominis plane block; US-OSTAP = ultrasound-guided oblique subcostal transversus abdominis plane; TAP block = transversus abdominis plane block; OSTAP = oblique subcostal transversus abdominis; US-STA = subcostal ultrasound transversus abdominis; US-Posterior TAP = ultrasound-guided posterior transversus abdominis plane block; Subcostal TAP block: subcostal transversus abdominis plane block; US-ESP = ultrasound-guided erector spinae plane; US-STA block = ultrasound-guided subcostal transversus abdominis; LAP-TAP = laparoscopic transversus abdominis plane; US-TAPB = ultrasound-guided transversus abdominis plane block; RSB = rectus sheath block; LAI = local anesthetic infiltration; TAPB = transversus abdominis plane block; IPLA = intraperitoneal local anesthetic injection; Modified BRILMA block = blocking the branches of intercostal nerves at the level of mid-axillary line; LSTAP = laparoscopic subcostal TAP; LAI-Group = received preoperational administration of 0.5% Ropivacaine plus 1 mcg of Dexamethasone at the trocar entrance; TL-Group = received preoperational administration of 0.5% Ropivacaine plus 1 mcg of Dexamethasone at the trocar entrance alongside posterior US-TAP block; TR-Group = US-TAP block combined with rectus sheath block.

**Table 3 jcm-11-06896-t003:** Adverse reactions, side effects, and recorded complications associated with local anesthetics used in laparoscopic cholecystectomy procedures.

No	Author/Year	Side Effects	Adverse Events	Complications	Drugs Used	Dose in mL or mg/kg
1	Ra (2010) [7]					
	US-TAP block (0.25%)	Sleep disturbance (*n* = 2)			Levobupivacaine 0.25%	30 mL
	US-TAP block 0.5%	Sleep disturbance (*n* = 0)			Levobupivacaine 0.5%	30 mL
	Control	Sleep disturbance (*n* = 6)				
2	Petersen (2012) [21]					
	US-TAP block (Ropivacaine)	Nausea scores 0–24 h (*n* = 0), with no difference in sedation scores.			Ropivacaine 0.5% + 2 mL of normal saline	22 mL
	US-TAP block (saline)	Nausea scores 0–24 h (*n* = 0), with no difference in sedation scores.			Ropivacaine 0.375%	20 mL
3	Shin (2014) [61]				Ropivacaine 0.375%	40 mL
	US-OSTAP block	Nausea: none (*n* = 15), mild (*n* = 0), moderate (*n* = 0), severe (*n* = 0), and shoulders pain (*n* = 2).				
	US-TAP block	Nausea: none (*n* = 12), mild (*n* = 2), moderate (*n* = 1), severe (*n* = 0), and shoulders pain (*n* = 0).				
	Control	Nausea: none (*n* = 11), mild (*n* = 1), moderate (*n* = 3), severe (*n* = 0), and shoulders pain (*n* = 1).				
4	Huang (2016) [67]					
	Control		Nausea (*n* = 3, vomiting *n* = 2, and abnormal sedation *n* = 2)			
	US-TAP Block, bilateral		Nausea (*n* = 1, vomiting *n* = 0, and abnormal sedation *n* = 0)		Ropivacaine 0.375%	30 mL
	US-TAP block + 2 mL of Dexamethasone		Nausea (*n* = 0, vomiting *n* = 0, and abnormal sedation *n* = 0)		Ropivacaine 0.375%	32 mL
5	Choi (2017) [71]					
	US-TAP block (indwelling catheter inserted)		Nausea (*n* = 11), vomiting (*n* = 2), dizziness (*n* = 2), headache (*n* = 0), urinary retention (*n* = 11), pain at the needle insertion site (*n* = 0), and hematoma (*n* = 0).		Ropivacaine 0.2%	20 mL
	US-TAP block + PCA		Nausea (*n* = 15), vomiting (*n* = 2), dizziness (*n* = 1), headache (*n* = 3), urinary retention (*n* = 3), pain at the needle insertion site (*n* = 0), and hematoma (*n* = 1).		Ropivacaine 0.2%	20 mL
	Control (PCA only)		Nausea (*n* = 9), vomiting (*n* = 2), dizziness (*n* = 2), headache (*n* = 1), urinary retention (*n* = 0), pain at the needle insertion site (*n* = 2), and hematoma (*n* = 1).		100 mL of normal saline + 40 mg Oxycodone and 180 mg of Ketorolac	
6	Houben (2019) [79]					
	US-TAP block (Levobupivacaine)	Fatigue median data (1 h *n* = 5, 2 h *n* = 5, 4 h *n* = 4.5, and 24 h = 4). Nausea median data (1 h *n* = 1, 2 h *n* = 0, 4 h *n* = 0, and 24 h = 0).			Levobupivacaine 0.375% + Epinephrine 5 mcg/mL	40 mL
	US-TAP block (saline)	Fatigue median data (1 h *n* = 5, 2 h *n* = 5, 4 h *n* = 3, and 24 h = 4). Nausea median data (1 h *n* = 0, 2 h *n* = 0, 4 h *n* = 0, and 24 h = 0).			40 mL 0.9% normal saline + Epinephrine 5 mcg/mL	40 mL
7	Janjua (2019) [80]					
	US-TAP block, unilateral			Respiratory depression (7.89%); others unclear	Bupivacaine 0.25%	0.4 mL/kg
	Control (port site infiltration)			Respiratory depression (2.56%); others unclear	Bupivacaine 0.25%	0.4 mL/kg
8	Siriwardana (2019) [84]					
	LAP-TAP + port site infiltration (× 4)	Vomiting episodes 0(0–4)			Bupivacaine 0.25%	40 mL + 12 − 20 mL
	Control (port site infiltration × 4)	Vomiting episodes 0(0–2)			Bupivacaine 0.25%	12–20 mL
9	Liang (2020) [88]					
	Group H	Postoperative nausea and vomiting were not significantly different between the 4 groups at 24 h (*p* = 0.180, *p* = 0.644).			Ropivacaine 0.75%	20 mL
	Group M				Ropivacaine 0.5%	20 mL
	Group L				Ropivacaine 0.2%	20 mL
	Group C				Normal saline 0.9%	20 mL
10	Ergin (2021) [90]					
	LAI Group			39 (97.5%)	Bupivacaine 0.5%	20 mL
	TAPB Group			40 (100%)	Bupivacaine 0.5% + 20 cc of physiologic saline	40 mL (20 + 20)
	IPLA Group			39 (97.5%)	Bupivacaine 0.5%	20 mL
	Control			40 (100%)		
11	Jung (2021) [91]					
	BD-TAP block, bilateral	Nausea (*n* = 4), and desaturation (*n* = 3).			Ropivacaine 0.25%	60 mL
	Control (sham block), bilateral	Nausea (*n* = 7), and desaturation (*n* = 2).			Normal saline 0.9%	60 mL
12	Han (2022) [98]					
	US-TAP block		Nausea and vomiting (*n* = 1), skin itching (*n* = 0), dizziness (*n* = 0), respiratory depression (*n* = 1), and puncture site hematoma (*n* = 0).		Ropivacaine 0.4% + 10 mg Tropisetron + 100 mL normal saline	142 mL
	Group S				Sufentanil 2 mg/kg via PCA + 10 mg Tropisetron + 100 mL normal saline	100 mL
	Group N		Nausea and vomiting (n = 8), skin itching (*n* = 1), dizziness (*n* = 0), respiratory depression (*n* = 2), and puncture site hematoma (*n* = 0).		Nalbuphine 2 mg/kg via PCA + 10 mg Tropisetron + 100 mL normal saline	100 mL
13	Lee (2022) [99]					
	US-TAP block (Ropivacaine)	1 h: nausea (*n* = 5), vomiting (*n* = 0); 8 h: nausea (*n* = 3), vomiting (*n* = 0); 24 h: nausea (*n* = 0), vomiting (*n* = 0).			Ropivacaine 0.375%	40 mL
	US-TAP block (normal saline)	1 h: nausea (*n* = 12), vomiting (*n* = 1); 8 h: nausea (*n* = 8), vomiting (*n* = 2); 24 h: nausea (*n* = 3), vomiting (*n* = 0).			Normal saline 0.9%	40 mL
14	Paudel (2022) [101]					
	US-TAP block	Nausea (*n* = 0), and vomiting (*n* = 0).			Bupivacaine 0.25%	40 mL
	Control (port site infiltration)	Nausea (*n* = 1), and vomiting (*n* = 2).			Bupivacaine 0.25%	20 mL

TAI Group = local anesthetic infiltration; TAPB Group = transversus abdominis plane block; IPLA Group = intraperitoneal local anesthetic injection; PCA = patient-controlled analgesia; US-TAP = ultrasound-guided transversus abdominis plane.

**Table 4 jcm-11-06896-t004:** Clinical significance of TAP.

No	Author (Year)	Drugs Used	Mean_post_	Mean_pre_	SDpre	Effect Size Index
1	Ortiz (2012) [20]	Morphine 24 h				
	US-TAP block		16.1	1.5	1.8	8.1
	Port infiltration		15.4	0.9	2.0	7.3

## Data Availability

The data that support the findings of this study are available from the corresponding author upon reasonable request.

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
