# Peer review of "Transversus Abdominis Plane Block as a Strategy for Effective Pain Management in Patients with Pain during Laparoscopic Cholecystectomy: A Systematic Review"

_jcm, 2022, doi:10.3390/jcm11236896_

Round 1
Reviewer 1 Report
This is a very thorough systematic review of an important subject. There are only very minor formatting issues eg. extra indent line 107.
It is a very long paper to read and I wonder if it could be separated into the paper itself and an appendix of some of the data.
Author Response
Response to Reviewer 1 Comments

Reviewer 2 Report
Review Comments: This paper reviews the use of TAP technology as a preoperative or postoperative analgesic protocol for patients undergoing cholecystectomy, and discusses different block points, different block onset time, different concentrations of local anesthetics and different combinations of adjuvant drugs, which has certain clinical guiding significance.
However, this paper has the following problems:
1. The results of the literature mentioned in this paper were not presented in the original paper (as in line 239-241 cited in reference No. 58), there are many incorrect dosages of anesthetic drugs and opioids in the text (e.g., item 17 in table3, remifentanil 15mg/kg; 337 lines of 2mg/kg sulfentanil, etc.), and the type and dose of opioids used are not clearly described in many places (such as lines 580-581, specific doses of 2ml dexmedetomidine?; The list goes on).
2. Regarding degree of pain in cholecystectomy, there are literatures supporting that this kind of pain is moderate or above, please complete and quote in the introduction.
3. In the column of table 3, named ‘anesthesia/analgesia‘, regarding intraoperative medication and analgesia should be divided into at least two columns.
4. Please clarify whether the intraoperative and postoperative opioids is changed to morphine equivalent, and whether the pain rescue opioids can be uniformly changed to morphine equivalent in the articles involved.
5. Considering the comparison of different concentrations of local anesthetic drugs used in the articles, it is recommended to include information on whether local anesthetic intoxication has occurred.
6. The ASA grade and BMI in the baseline data of patients have an impact on the patient's local anesthetic tolerance. Please add such information.
Author Response
Response to Reviewer 2 comments

Reviewer 3 Report
Thank you for the possibility to review the article entitled “Transversus abdominis block (TAP) as a strategy for effective pain management in patients with pain during laparoscopic cholecystectomy: a systematic review”.
I have some remarks:
- I personally prefer a more focused introduction, leaving the literature overlook for the discussion section. However, it is acceptable in the present form.
- Search: transversus abdominis plane block was not captured in your search. Instead, you used transversus abdominis block, transversus abdominal or transverse abdominis. This is a limitation as in the literature “transversus abdominis plane block” is the most used terminology. I also suggest using * to better search the terms. E.g. “transvers*”AND”abdomin*”AND”block” should capture more precisely all articles needed. Beyond PubMed and WOS, this search method can be also applied to other medical databases.
- Results: you stated that most patients were female, however, they account for 49.9% of patients.
- Table 4 is too large to be presented in a manuscript. I suggest presenting it as supplementary appendix.
- In my opinion, the most relevant study limitation is related to the wide inclusion criteria and general objectives. Consequently, some objectives (e.g., Objective #3) are presented only descriptively and no analysis was carried out. The interest of readers is regarding the optimal dose of anesthetics and the medications used. Similarly, no quantitative analysis was carried out regarding which TAP block technique should be used. The literature is consistent regarding the effectiveness of TAP block in laparoscopic procedures, however, how should we do it, is less known. Unfortunately, this systematic review, although a huge work was done by the authors, does not answer to all relevant questions regarding the TAP block in LC. A qualitative analysis without any quantitative analysis on a so wide topic does not significantly improve the actual body of evidence.
Author Response
Response to Reviewer 3 comments

Reviewer 4 Report
Author attempted this comprehension review about the Transversus abdominis block (TAP) as a strategy for effective pain management in patients with pain during laparoscopic cholecystectomy. However, there are few concerns that need to be addressed.
1. In table 1, the results needs to be focused on the significant outcomes, that would add more scientific weightage and increase the audience visibility. If necessary, the table can be polished further.
2. Page3, Line 100: In the inclusion criteria, it was mentioned the adult patients undergoing LC were included. However, in Line 190, it was mentioned the age of study subjects ranging from 18-80 years, please clarify in the text.
3. Page 1, Line 7: Correct the typo error in the country name "Soudi Arabia".
Author Response
Response to reviewer 4 comments

Round 2
Reviewer 2 Report
Some adjustments have made to the previous suggestions, but still some problems to further optimize:
1. At the beginning of the article, the problems to be solved have been listed, and it is recommended to summarize the answers to these 5 questions in the conclusion.
2. In the result presentation part, the content of each sub-point is little chaotic, and maybe it can be divided into two parts, ultrasound-guided and laparoscopic guided according to the different ways of TAP, and then divided into sub-points to make the display of results clearer. As a relatively new technology, US-OSTAP is essentially consistent with US-TAP, so advised to be discussed as a sub-point under US-TAP
3. Lines 543-550 have nothing to do with the sub-point. Please make adjustments and the author should review the full text carefully to avoid repeating such mistakes.
Author Response
Please find attached the response to reviewer 2 comments

Reviewer 3 Report
Some comments were addressed. I have no further remark.
Author Response
We have checked the manuscript again
(supplemental material inclusive)
for grammar/spelling errors and corrected them.